# Robust $k$-means: a Theoretical Revisit

**Alexandros Georgogiannis**
School of Electrical and Computer Engineering
Technical University of Crete, Greece
alexandrosgeorgogiannis at gmail.com

## Abstract

Over the last years, many variations of the quadratic $k$-means clustering procedure have been proposed, all aiming to robustify the performance of the algorithm in the presence of outliers. In general terms, two main approaches have been developed: one based on penalized regularization methods, and one based on trimming functions. In this work, we present a theoretical analysis of the robustness and consistency properties of a variant of the classical quadratic $k$-means algorithm, the robust $k$-means, which borrows ideas from outlier detection in regression. We show that two outliers in a dataset are enough to breakdown this clustering procedure. However, if we focus on "well-structured" datasets, then robust $k$-means can recover the underlying cluster structure in spite of the outliers. Finally, we show that, with slight modifications, the most general non-asymptotic results for consistency of quadratic $k$-means remain valid for this robust variant.

## 1 Introduction

Let $\phi : \mathbb{R} \to \mathbb{R}_+$ be a lower semi-continuous (lsc) and symmetric function with minimum value $\phi(0)$. Given a set of points $\mathcal{X}^n = \{x_1, \ldots, x_n\} \subset \mathbb{R}^p$, consider the generalized $k$-means problem (GKM) [7]

$$\min_{c_1, \ldots, c_k} R_n(c_1, \ldots, c_k) = \sum_{i=1}^{n} \min_{1 \leq l \leq k} \phi(||x_i - c_l||_2) \qquad \text{(GKM)}$$

$$\text{subject to} \quad c_l \in \mathbb{R}^p, \ l \in \{1, \ldots, k\}.$$

Our aim is to find a set of $k$ centers $\{c_1, \ldots, c_k\}$ that minimize the clustering risk $R_n$. These centers define a partition of $\mathcal{X}^n$ into $k$ clusters $\mathcal{A} = \{A_1, \ldots, A_k\}$, defined as

$$A_l = \left\{ x \in \mathcal{X}^n : l = \operatorname{argmin}_{1 \leq j \leq k} \phi(||x - c_j||_2) \right\}, \qquad (1)$$

where ties are broken randomly. Varying $\phi$ beyond the usual quadratic function ($\phi(t) = t^2$) we expect to gain some robustness against the outliers [9]. When $\phi$ is upper bounded by $\delta$, the clusters are defined as follows. For $l \leq k$, let

$$A_l = \left\{ x \in \mathcal{X}^n : l = \operatorname{argmin}_{1 \leq j \leq k} \phi(||x - c_j||_2) \text{ and } \phi(||x - c_l||_2) \leq \delta \right\}, \qquad (2)$$

and define the extra cluster

$$A_{k+1} = \left\{ x \in \mathcal{X}^n : \min_{1 \leq j \leq k} \phi(||x - c_j||_2) > \delta \right\}. \qquad (3)$$

This extra cluster contains points whose distance from their closest center, when measured according to $\phi(||x - c_l||_2)$, is larger than $\delta$ and, as will become clear later, it represents the set of outliers. From now on, given a set of centers $\{c_1, \ldots, c_k\}$, we write just $\mathcal{A} = \{A_1, \ldots, A_k\}$ and implicitly mean $\mathcal{A} \cup A_{k+1}$ when $\phi$ is bounded.[1]

Now, consider the following instance of (GKM), for the same set of points $\mathcal{X}^n$,

$$\min_{c_1,\ldots,c_k} \ R'_n(c_1,\ldots,c_k) = \sum_{i=1}^{n} \min_{1 \le l \le k} \left\{ \underbrace{\min_{o_i} \frac{1}{2}\|x_i - c_l - o_i\|_2^2 + f_\lambda(\|o_i\|_2)}_{\phi(\|x_i - c_l\|_2)} \right\} \quad \text{(RKM)}$$

$$\text{subject to } c_l \in \mathbb{R}^p, l = 1,\ldots,k,$$
$$o_i \in \mathbb{R}^p, i = 1,\ldots,n,$$

where $f_\lambda : \mathbb{R} \to \mathbb{R}_+$ is a symmetric, lsc, proper[2] and bounded from below function, with minimum value $f_\lambda(0)$, and $\lambda$ a non-negative parameter. This problem is called robust $k$-means (RKM) and, as we show later, it takes the form of (GKM) when $\phi$ equals the Moreau envelope of $f_\lambda$. The problem (RKM) [5, 24] describes the following simple model: we allow each observation $x_i$ to take on an "error" term $o_i$ and we penalize the errors, using a group penalty, in order to encourage most of the observations' errors to be equal to zero. We consider functions $f_\lambda$ where the parameter $\lambda \ge 0$ has the following effect: for $\lambda = 0$, all $o_i$'s may become arbitrary large (all observations are outliers), while, for $\lambda \to \infty$, all $o_i$'s become zero (no outliers); non-trivial cases occur for intermediate values $0 < \lambda < \infty$. Our interest is in understanding the robustness and consistency properties of (RKM).

**Robustness:** Although robustness is an important notion, it has not been given a standard technical definition in the literature. Here, we focus on the *finite sample breakdown point* [18], which counts how many outliers a dataset may contain without causing significant damage in the estimates of the centers. Such damage is reflected to an arbitrarily large magnitude of at least one center. In Section 3, we show that two outliers in a dataset are enough to breakdown some centers. On the other hand, if we restrict our focus on some "well structured" datasets, then (RKM) has some remarkable robustness properties even if there is a considerable amount of contamination.

**Consistency:** Much is known about the consistency of (GKM) when the function $\phi$ is lsc and increasing [11, 15]. It turns out that this case also includes the case of (RKM) when $f_\lambda$ is convex (see Section 3.1 for details). In Section 4, we show that the known non-asymptotic results about consistency of quadratic $k$-means may remain valid even when $f_\lambda$ is non-convex.

## 2 Preliminaries and some technical remarks

We start our analysis with a few technical tools from variational analysis [19]. Here, we introduce the necessary notation and a lemma (the proofs are in the appendix). The Moreau envelope $e_f^\mu(x)$ with parameter $\mu > 0$ (Definition 1.22 in [19]) of an lsc, proper, and bounded from below function $f : \mathbb{R}^p \to \mathbb{R}$ and the associated (possibly multivalued) proximal map $P_f^\mu : \mathbb{R}^p \rightrightarrows \mathbb{R}^p$ are

$$e_f^\mu(x) = \min_{z \in \mathbb{R}^p} \frac{1}{2\mu}\|x - z\|_2^2 + f(z) \ \text{ and } \ P_f^\mu(x) = \operatorname{argmin}_{z \in \mathbb{R}^p} \frac{1}{2\mu}\|x - z\|_2^2 + f(z), \quad (4)$$

respectively. In order to simplify the notation, in the following, we fix $\mu$ to 1 and suppress the superscript. The Moreau envelope is a continuous approximation from below of $f$ having the same set of minimizers while the proximal map gives the (possibly non-unique) minimizing arguments in (4). For (GKM), we define $\Phi : \mathbb{R}^p \to \mathbb{R}$ as $\Phi(x) := \phi(\|x\|_2)$. Accordingly, for (RKM), we define $F_\lambda : \mathbb{R}^p \to \mathbb{R}$ as $F_\lambda(x) := f_\lambda(\|x\|_2)$. Thus, we obtain the following pairs:

$$e_{f_\lambda}(x) := \min_{o \in \mathbb{R}} \frac{1}{2}(x-o)^2 + f_\lambda(o), \quad P_{f_\lambda}(x) := \operatorname{argmin}_{o \in \mathbb{R}} e_{f_\lambda}(x), \ x \in \mathbb{R} \quad (5a)$$

$$e_{F_\lambda}(x) := \min_{o \in \mathbb{R}^p} \frac{1}{2}\|x-o\|_2^2 + F_\lambda(o), \quad P_{F_\lambda}(x) := \operatorname{argmin}_{o \in \mathbb{R}^p} e_{F_\lambda}(x), \ x \in \mathbb{R}^p. \quad (5b)$$

Obviously, (RKM) is equivalent to (GKM) when $\Phi(x) = e_{F_\lambda}(x)$. Every map $\mathtt{P} : \mathbb{R} \rightrightarrows \mathbb{R}$ throughout the text is assumed to be $i)$ odd, i.e., $\mathtt{P}(-x) = -\mathtt{P}(x)$, $ii)$ compact-valued, $iii)$ non-decreasing, and $iv)$ have a closed graph. We know that for any such map there exists at least one function $f_\lambda$ such that $\mathtt{P} = P_{f_\lambda}$ (Proposition 3 in [26]).[3] Finally, for our purposes (outlier detection), it is natural

to require that $v$) P is a *shrinkage rule*, i.e., $P(x) \le x, \forall x \ge 0$. The following corollary is quite straightforward and useful in the sequel.

**Corollary 1.** *Using the notation in definitions (5a) and (5b), we have*

$$P_{F_\lambda}(x) = \frac{x}{||x||_2} P_{f_\lambda}(||x||_2) \ \ and \ \ e_{F_\lambda}(x) = e_{f_\lambda}(||x||_2). \tag{6}$$

Passing from a model of minimization in terms of a single problem, like (GKM), to a model in which a problem is expressed in a particular parametric form, like (RKM) with the Moreau envelope, the description of optimality conditions is opened to the incorporation of the multivalued map $P_{F_\lambda}$. The next lemma describes the necessary conditions for a center $c_l$ to be (local) optimal for (RKM). Since we deal with the general case, well known results, such as smoothness of the Moreau envelope or convexity of its subgradients, can no longer be taken for granted.

**Remark 1.** *Let $\Phi(\cdot) = e_{F_\lambda}(\cdot)$. The usual subgradient, denoted as $\hat{\partial}\Phi(x)$, is not sufficient to characterize the differentiability properties of $R'_n$ in (RKM). Instead, we use the (generalized) subdifferential $\partial\Phi(x)$ (Definition 8.3 in [19]). For all $x$, we have $\hat{\partial}\Phi(x) \subseteq \partial\Phi(x)$. Usually, the previous two sets coincide at a point $x$. In this case, $\Phi$ is called regular at $x$. However, it is common in practice that the sets $\hat{\partial}\Phi(x)$ and $\partial\Phi(x)$ are different (for a detailed exposition on subgradients see Chapter 8 in [19]; see also Example 1 in Appendix A.9).*

**Lemma 1.** *Let $P_{F_\lambda} : \mathbb{R}^p \rightrightarrows \mathbb{R}^p$ be a proximal map and set $\Phi(\cdot) = e_{F_\lambda}(\cdot)$. The necessary (generalized) first order conditions for the centers $\{c_1, \ldots, c_k\} \subset \mathbb{R}^p$ to be optimal for (RKM) are*

$$0 \in \partial\Big\{ \sum_{i \in A_l} \Phi(x_i - c_l) \Big\} \subseteq \sum_{i \in A_l} \partial\Phi(x_i - c_l) \subseteq \sum_{i \in A_l} \left( c_l - x_i + P_{F_\lambda}(x_i - c_l) \right), \ l \in \{1, \ldots, k\}. \tag{7}$$

The interpretation of the set inclusion above is the following: for any center $c_l \in \mathbb{R}^p$, every subgradient vector in $\partial\Phi(x_i - c_l)$ *must be a vector associated with a vector in* $P_{F_\lambda}(x_i - c_l)$ (Theorem 10.13 in [19]). However, in general, the converse does not hold true. We note that when the proximal map is single-valued and continuous, which happens for example not only when $f_\lambda$ is convex, but also for many popular non-convex penalties, both set inclusions become equalities and the converse holds, i.e., every vector in $P_{F_\lambda}(x_i - c_l)$ *is a vector associated with a subgradient in* $\partial\Phi(x_i - c_l)$ (Theorem 10.13 in [19] and Proposition 7 in [26]).

We close this section with some useful remarks on the properties of the Moreau envelope as a map between two spaces of functions. There exist cases where two different functions, $f_\lambda \ne f'_\lambda$, have equal Moreau envelopes, $e_{f_\lambda} = e_{f'_\lambda}$ (Proposition 1 in [26]), implying that two different forms of (RKM) correspond to the same $\phi$ in (GKM). For example, the proximal hull of $f_\lambda$, defined as $h^\mu_{f_\lambda}(x) := -e^\mu_{(-e^\mu_{f_\lambda})}(x)$, is a function different from $f_\lambda$ but has the same Moreau envelope as $f_\lambda$ (see also Example 1.44 in [19], Proposition 2 and Example 3 in [26]). This is the main reason we preferred the proximal map instead of the penalty function point of view for the analysis of (RKM).

# 3 On the breakdown properties of robust $k$-means

In this section, we study the finite sample breakdown point of (RKM) and, more specifically, its universal breakdown point. Loosely speaking, the breakdown point measures the minimum fraction of outliers that can cause excessive damage in the estimates of the centers. Here, it will become clear how the interplay between the two forms, (GKM) and (RKM), helps the analysis. Given a dataset $\mathcal{X}^n = \{x_1, \ldots, x_n\}$ and a nonnegative integer $m \le n$, we say that $\mathcal{X}^n_m$ is an $m$-modification if it arises from $\mathcal{X}^n$ after replacing $m$ of its elements by arbitrary elements $x'_i \in \mathbb{R}^p$ [6]. Denote as $r(\lambda)$ the non-outlier samples, as counted after solving (RKM), for a dataset $\mathcal{X}^n$ and some $\lambda \ge 0$, i.e., [4]

$$r(\lambda) := \Big| \{ x_i \in \mathcal{X}^n \ : \ ||o_i||_2 = 0, \ i = 1, \ldots, n \} \Big|. \tag{8}$$

Then, the number of estimated outliers is $q(\lambda) = n - r(\lambda)$. In order to simplify notation, we drop the dependence of $r$ and $q$ on $\lambda$. With this notation, we proceed to the following definition.

**Definition 1** (universal breakdown point for the centers [6]). *Let $n, r, k$ be such that $n \geq r \geq k+1$. Given a dataset $\mathcal{X}_m^n$ in $\mathbb{R}^p$, let $\{c_1, \ldots, c_k\}$ denote the (global) optimal set of centers for (RKM). The universal breakdown value of (RKM) is*

$$\beta(n, r, k) := \min_{\mathcal{X}^n} \min_{1 \leq m \leq n} \left\{ \frac{m}{n} : \sup_{\mathcal{X}_m^n} \max_{1 \leq l \leq k} ||c_l||_2 = \infty \right\}. \tag{9}$$

*Here, $\mathcal{X}^n = \{x_1, \ldots, x_n\} \subset \mathbb{R}^p$ while $\mathcal{X}_m^n \subset \mathbb{R}^p$ runs over all $m$-modifications of $\mathcal{X}^n$.*

According to the concept of universal breakdown point, (RKM) breaks down at the first integer $m$ for which *there exists* a set $\mathcal{X}^n$ such that the estimates of the cluster centers become arbitrarily bad for a *suitable modification $\mathcal{X}_m^n$*. Our analysis is based on $P_{f_\lambda}$ and considers two cases: those of biased and unbiased proximal maps. The former corresponds to the class of convex functions $f_\lambda$, while the latter corresponds to a class of non-convex $f_\lambda$.

### 3.1 Biased proximal maps: the case of convex $f_\lambda$

If $f_\lambda$ is convex, then $\Phi = e_{F_\lambda}$ is also convex while $P_{F_\lambda}$ is continuous, single-valued, and satisfies [19]

$$||x - P_{F_\lambda}(x)||_2 \to \infty \quad \text{as} \quad ||x||_2 \to \infty. \tag{10}$$

Proximal maps with this property are called *biased* since, as the $l_2$-norm of $x$ increases, so does the norm of the difference in (10). In this case, for each $x_i \in A_l$, from Lemma 1 and expression (10), we have

$$||\nabla\Phi(x_i - c_l)||_2 = ||\nabla e_{F_\lambda}(x_i - c_l)||_2 = ||c_l - x_i + P_{F_\lambda}(x_i - c_l)||_2 \to \infty \text{ as } ||x_i - c_l||_2 \to \infty. \tag{11}$$

The supremum value of $||\nabla\Phi(x - c_l)||_2$ is closely related to the *gross error sensitivity* of an estimator [9]. It is interpreted as the worst possible influence which a sample $x$ can have on $c_l$ [7]. In view of (11) and the definition of the clusters in (1), (RKM) is extremely sensitive. Although it can detect an outlier, i.e., a sample $x_i$ with a nonzero estimate for $||o_i||_2$, it *does not reject it* since the influence of $x_i$ on its closest center never vanishes.[5] The $l_1$-norm, $f_\lambda(x) = \lambda|x|$, which has Moreau envelope equal to the Huber loss-function [24], is the limiting case for the class of convex penalty functions that, although it keeps the difference $||x - P_{F_\lambda}(x)||_2$ in (10) constant and equal to $\lambda$, introduces a bias term proportional to $\lambda$ in the estimate $c_l$. The following proposition shows that (RKM) with a biased $P_{F_\lambda}$ has breakdown point equal to $\frac{1}{n}$, i.e., one outlier suffices to breakdown a center.

**Proposition 1.** *Assume $k \geq 2$, $k+1 < r \leq n$. Given a biased proximal map, there exist a dataset $\mathcal{X}^n$ and a modification $\mathcal{X}_1^n$ such that (RKM) breaks down.*

### 3.2 Unbiased proximal maps: the case of non-convex $f_\lambda$

Consider now the $l_0$-(pseudo)norm on $\mathbb{R}$, $f_\lambda(z) := \lambda|z|_0 = \frac{\lambda^2}{2}\mathbb{1}_{\{z \neq 0\}}$, and the associated hard-thresholding proximal operator $P_{\lambda|\cdot|_0} : \mathbb{R} \rightrightarrows \mathbb{R}$,

$$P_{\lambda|\cdot|_0}(x) = \arg\min_{z \in \mathbb{R}} \tfrac{1}{2}(x - z)^2 + f_\lambda(z) = \begin{cases} 0, & |x| < \lambda, \\ \{0, x\}, & |x| = \lambda, \\ x, & |x| > \lambda. \end{cases} \tag{12}$$

According to Lemma 1, for $p = 1$ (scalar case), we have

$$\partial\Phi(x_i - c_l) \subseteq c_l - x_i + P_{\lambda|\cdot|_0}(x_i - c_l) \overset{(12)}{=} \{0\} \text{ for } |x_i - c_l| > \lambda, \; x_i \in A_l, \tag{13}$$

implying that $\Phi(x_i - c_l)$, as a function of $c_l$, remains constant for $|x_i - c_l| > \lambda$. As a consequence of (13), if $c_l$ is local optimal, then $0 \in \partial\{\sum_{i \in A_l} \Phi(x_i - c_l)\}$ and

$$0 \in \sum_{\substack{i \in A_l, \\ |x_i - c_l| < \lambda}} (c_l - x_i) + \sum_{\substack{i \in A_l, \\ |x_i - c_l| = \lambda}} \left( c_l - x_i + P_{\lambda|\cdot|}(x_i - c_l) \right). \tag{14}$$

Depending on the value of $\lambda$, (RKM) with the $l_0$-norm is able to ignore samples with distance from their closest center larger than $\lambda$. This is done since $P_{\lambda|\cdot|_0}(x_i - c_l) = x_i - c_l$ whenever $|x_i - c_l| > \lambda$

and the influence of $x_i$ vanishes. In fact, there is a whole family of non-convex $f_\lambda$'s whose proximal map $P_{f_\lambda}$ satisfies

$$P_{f_\lambda}(x) = x, \ \text{ for all } \ |x| > \tau,\qquad(15)$$

for some $\tau > 0$. These are called *unbiased* proximal maps [13, 20] and have the useful property that, as one observation is arbitrarily modified, all estimated cluster centers remain bounded by a constant that depends only on the remaining unmodified samples. Under certain circumstances, the proof of the following proposition reveals that, if there exists one outlier in the dataset, then robust $k$-means will reject it.

**Proposition 2.** *Assume $k \geq 2$, $k + 1 < r \leq n$, and consider the dataset $\mathcal{X}^n = \{x_1, \ldots, x_n\}$ along with its modification by one replacement $y$, $\mathcal{X}_1^n = \{x_1, \ldots, x_{n-1}, y\}$. If we solve (RKM) with $\mathcal{X}_1^n$ and an unbiased proximal map satisfying (15), then all estimates for the cluster centers remain bounded by a constant that depends only on the unmodified samples of $\mathcal{X}^n$.*

Next, we show that, even for this class of maps, there always exists a dataset that causes one of the estimated centers to breakdown as two particular observations are suitably replaced.

**Theorem 1** (Universal breakdown point for (RKM)). *Assume $k \geq 2$ and $n \geq r \geq k + 2$. Given an unbiased proximal map satisfying (15), there exist a dataset $\mathcal{X}^n$ and a modification $\mathcal{X}_2^n$, such that (RKM) breaks down.*

Hence, the universal breakdown point of (RKM) with an unbiased proximal map is $\frac{2}{n}$. In Figure 1, we give a visual interpretation of Theorem 1. The top subfigure depicts the unmodified initial dataset $\mathcal{X}^9 = \{x_1, \ldots, x_9\}$ (black circles) with a clear two-cluster structure; the bottom subfigure shows the modification $\mathcal{X}_2^9$ (dashed line arrows). Theorem 1 states that (RKM) on $\mathcal{X}_2^9$ fails to be robust since, every subset of $\mathcal{X}_2^9$ with $r = 8$ points has a cluster containing an outlier.

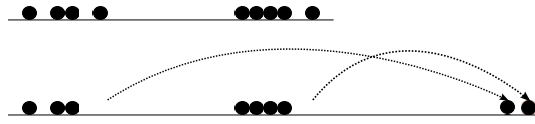

Figure 1: The top subfigure is the unmodified dataset $\mathcal{X}^9$. Theorem 1 states that every subset of the modification $\mathcal{X}_2^9$ (bottom subfigure) with size 8 contains an outlier.

### 3.3 Restricted robustness of robust $k$-means for well-clustered data

The result of Theorem 1 is disappointing but it is not (RKM) to be blamed for the poor performance but the tight notion of the definition about the breakdown point [6, 7]; allowing any kind of contamination in a dataset is a very general assumption.

In this section, we place two restrictions: i) we consider datasets where inlier samples can be covered by unions of balls with centers that are "far apart" each other, and ii) we ask a question different from the finite sample breakdown point. We want to exploit as much as possible the results of [2] concerning a new quantitative measure of noise robustness which compares the output of (RKM) on a contaminated dataset to its output on the uncontaminated version of the dataset. Our aim is to show that (RKM), with a certain class of proximal maps and datasets that are well-structured *ignores the influence of outliers* when grouping the inliers.

First, we introduce Corollary 2 which states the form that $P_{f_\lambda}$ should have in order the results of [2] to apply to (RKM) and, second, we give details about the datasets which we consider as well-structured. Using this corollary we are able to design proximal maps for which Theorems 3, 4, and 5 in [2] apply; otherwise, it is not clear how the analysis of [2] is valid for (RKM).

Let $h : \mathbb{R} \to \mathbb{R}$ be a continuous function with the following properties:

1. $h$ is odd and non-decreasing ($h_+(\cdot)$ is used to denote its restriction on $[0, \infty)$);

2. $h$ is a shrinkage rule: $0 \leq h_+(x) \leq x$, $\forall x \in [0, \infty)$;

3. the difference $x - h_+(x)$ is non-decreasing, i.e., for $0 \leq x_1 \leq x_2$ we have $x_1 - h_+(x_1) \leq x_2 - h_+(x_2)$.

Define the map

$$P_{f_\lambda}(x) := \begin{cases} h(x), & |x| < \lambda, \\ \{h(x), x\}, & |x| = \lambda, \\ x, & |x| > \lambda. \end{cases} \qquad (16)$$

Multivaluedness of $P_{f_\lambda}$ at $|x| = \lambda$ signals that $e_{f_\lambda}$ is non-smooth at these points. An immediate consequence for the Moreau envelope associated with the previous map is the following.

**Corollary 2.** *Let the function $g : [0, \infty) \to [0, \infty)$ be defined as*

$$g(x) := \int_0^x (u - h(u))du, \ x \in [0, \infty). \qquad (17)$$

*Then, the Moreau envelope associated with $P_{f_\lambda}$ in (16) is*

$$e_{f_\lambda}(x) = \min\{g(|x|), g(\lambda)\} = g(\min\{|x|, \lambda\}). \qquad (18)$$

Next, we define what it means for a dataset to be $(\rho_1, \rho_2)$-*balanced*; this is the class of datasets which we consider to be *well-structured*.

**Definition 2** (($\rho_1, \rho_2$) balanced dataset [2]). *Assume that a set $\mathcal{X}^n \subset \mathbb{R}^p$ has a subset $\mathcal{I}$ (inliers), with at least $\frac{n}{2}$ samples, and the following properties:*

1. *$\mathcal{I} = \bigcup_{l=1}^k B_l$, where $B_l = B(b_l, r)$ is a ball in $\mathbb{R}^p$ with bounded radius $r$ and center $b_l$;*

2. *$\rho_1|\mathcal{I}| \le |B_l| \le \rho_2|\mathcal{I}|$ for every $l$, where $|B_l|$ is the number of samples in $B_l$ and $\rho_1, \rho_2 > 0$;*

3. *$||b_l - b_{l'}||_2 > v$ for every $l \ne l'$, i.e., the centers of the balls are at least $v > 0$ apart.*

*Then, $\mathcal{X}^n$ is a $(\rho_1, \rho_2)$-balanced dataset.*

We now state the form that Theorem 3 in [2] takes for (RKM).

**Theorem 2** (Restricted robustness of (RKM)). *If i) $e_{f_\lambda}$ is as in Corollary 2, i.e., $e_{f_\lambda}(||x||_2) = g(\min\{||x||_2, \lambda\})$, ii) $\mathcal{X}^n$ has a $(\rho_1, \rho_2)$-balanced subset of samples $\mathcal{I}$ with $k$ balls, and iii) the centers of the balls are at least $v > 4r + 2g^{-1}(\frac{\rho_1 + \rho_2}{\rho_1} g(r))$ apart, then for $\lambda \in \left[\frac{v}{2}, g^{-1}\left(\frac{|\mathcal{I}|}{|\mathcal{X}^n \setminus \mathcal{I}|}(\rho_1 g(\frac{v}{2} - 2r) - (\rho_1 + \rho_2)g(r))\right)\right]$ the set of outliers $\mathcal{X}^n \setminus \mathcal{I}$ has no effect on the grouping of inliers $\mathcal{I}$. In other words, if $\{x, y\} \in B_l$ and $\{c_1, \ldots, c_k\}$ are the optimal centers when solving (RKM) for a $\lambda$ as described before, then*

$$l = \mathrm{argmin}_{1 \le j \le k} e_{f_\lambda}(||x - c_j||_2) = \mathrm{argmin}_{1 \le j \le k} e_{f_\lambda}(||y - c_j||_2).$$

For the sake of completeness, we give a proof of this theorem in the appendix. In a similar way, we can recast the results of Theorems 4 and 5 in [2] to be valid for (RKM).

## 4   On the consistency of robust $k$-means

Let $\mathcal{X}^n$ be a set with $n$ independent and identically distributed random samples $x_i$ from a fixed but unknown probability distribution $\mu$. Let $\hat{\mathcal{C}}$ be the empirical optimal set of centers, i.e.,

$$\hat{\mathcal{C}} := \mathrm{argmin}_{c_1 \ldots, c_k \in \mathbb{R}^p} R'_n(c_1, \ldots, c_k). \qquad (19)$$

The *population* optimal set of centers is the set

$$\mathcal{C}^* := \mathrm{argmin}_{c_1 \ldots, c_k \in \mathbb{R}^p} R'(c_1, \ldots, c_k), \qquad (20)$$

where $R'$ is the population clustering risk, defined as

$$R'(c_1, \ldots, c_k) := \int \min_{1 \le l \le k} \left\{ \underbrace{\min_{o \in \mathbb{R}^p} \frac{1}{2}||x - c_l - o||_2^2 + f_\lambda(||o||_2)}_{\phi(||x - c_l||_2) = e_{f_\lambda}(||x - c_l||_2)} \right\} \mu(dx). \qquad (21)$$

Loss consistency and (simply) consistency for (RKM) require, respectively, that

$$R'_n(\hat{\mathcal{C}}) \overset{n \to \infty}{\Longrightarrow} R'(\mathcal{C}^*) \ \text{ and } \ \hat{\mathcal{C}} \overset{n \to \infty}{\Longrightarrow} \mathcal{C}^*. \qquad (22)$$

In words, as the size $n$ of the dataset $\mathcal{X}^n$ increases, the empirical clustering risk $R'_n(\hat{\mathcal{C}})$ converges almost surely to the minimum population risk $R'(\mathcal{C}^*)$ and (for $n$ large enough) $\hat{\mathcal{C}}$ can effectively replace the optimal set $\mathcal{C}^*$ in quantizing the unknown probability measure $\mu$.

For the case of convex $f_\lambda$, non-asymptotic results describing the rate of convergence of $R'_n$ to $R$ in (22) are already known ([11], Theorem 3). Noting that the Moreau envelope of a non-convex $f_\lambda$ belongs to a class of functions with polynomial discrimination [16] (the shatter coefficient of this class is bounded by a polynomial) we give a sketch proof of the following result.

**Theorem 3** (Consistency of (RKM)). *Let the samples $x_i \in \mathcal{X}^n$, $i \in \{1, \ldots, n\}$, come from a fixed but unknown probability measure $\mu$. For any $k \geq 1$ and any unbiased proximal map, we have*

$$\lim_{n \to \infty} \mathbb{E}R'(\hat{\mathcal{C}}) \to R'(\mathcal{C}^*) \quad and \quad \lim_{n \to \infty} \hat{\mathcal{C}} \to \mathcal{C}^* \;\; (convergence \; in \; probability). \tag{23}$$

Theorem 3 reads like an asymptotic convergence result. However, its proof (given in the appendix) uses combinatorial tools from Vapnik-Chervonenkis theory, revealing that the non-asymptotic rate of convergence of $\mathbb{E}R'(\hat{\mathcal{C}})$ to $R'(\mathcal{C}^*)$ is of order $O(\sqrt{\log n / n})$ (see Corollary 12.1 in [4]).

## 5   Relating (RKM) to trimmed $k$-means

As the effectiveness of robust $k$-means on real world and synthetic data has already been evaluated [5, 24], the purpose of this section is to relate (RKM) to trimmed k-means (TKM) [7]. Trimmed $k$-means is based on the methodology of "impartial trimming", which is a combinatorial problem fundamentally different from (RKM). Despite their differences, the experiments show that, both (RKM) and (TKM) perform remarkably similar in practice. The solution of (TKM) (which is also a set of $k$ centers) is the solution of quadratic $k$-means on the subsample containing $\lceil n(1 - \alpha)\rceil$ points with the smallest mean deviation $(0 < \alpha < 1)$. The only common characteristic of (RKM) and (TKM) is that they both have the same universal breakdown point, i.e., $\frac{2}{n}$, for arbitrary datasets.

Trimmed $k$-means takes as input a dataset $\mathcal{X}^n$, the number of clusters $k$, and a proportion of outliers $a \in (0, 1)$ to remove.[6] A popular heuristic algorithm for (TKM) is the following. After the initialization, each iteration of (TKM) consists of the following steps: $i)$ the distance of each observation from its closest center is computed, $ii)$ the top $\lceil an \rceil$ observations with larger distance from its closest center are removed, $iii)$ the remaining points are used to update the centers. The previous three steps are repeated untill the centers converge.[7] As for robust $k$-means, we solve the (RKM) problem with a coordinate optimization procedure (see Appendix A.9 for details).

The synthetic data for the experiments come from a mixture of Gaussians with 10 components and without any overlap between them.[8] The number of inlier samples is 500 and each inlier $x_i \in [-1, 1]^{10}$ for $i \in \{1, \ldots, 500\}$. On top of the inliers lie 150 outliers in $\mathbb{R}^{10}$ distributed uniformly in general positions over the entire space. We consider two scenarios: in the first, the outliers lie in $[-3, 3]^{10}$ (call it mild-contamination), while, in the second, the outliers lie in $[-6, 6]^{10}$ (call it heavy-contamination). The parameter $a$ in trimmed $k$-means (the percentage of outliers) is set to $a = 0.3$, while the value of the parameter $\lambda$ for which (RKM) yields 150 outliers is found through a search over a grid on the set $\lambda \in (0, \lambda_{\max})$ (we set $\lambda_{\max}$ as the maximum distance between two points in a dataset). Both algorithms, as they are designed, require as input an initial set of $k$ points; these points form the initial set of centers. In all experiments, both (RKM) and (TKM) take the same $k$ vectors as initial centers, i.e., $k$ points sampled randomly from the dataset.

The statistics we use for the comparison are: $i)$ the rand-index for clustering accuracy [17] $ii)$ the cluster estimation error, i.e., the root mean square error between the estimated cluster centers and the sample mean of each cluster, $iii)$ the true positive outlier detection rate, and finally, $iv)$ the false positive outlier detection rate. In Figures 2-3, we plot the results for a proximal map $P_f$ like the one in (16) with $h(x) = \alpha x$ and $\alpha = 0.005$; with this choice for $h$, we mimic the hard-thresholding operator. The results for each scenario (accuracy, cluster estimation error, etc) are averages over 150 runs of the experiment. As seen, both algorithms share almost the same statistics in all cases.

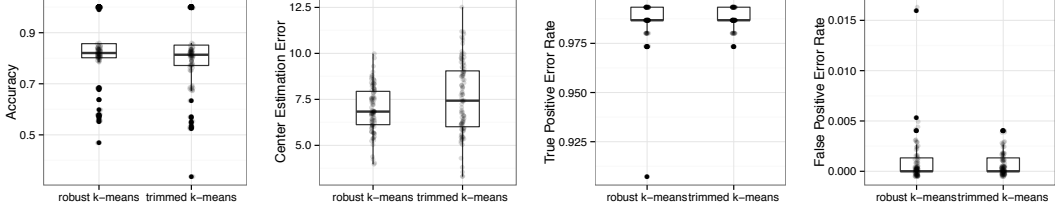

Figure 2: Performance of robust and trimmed $k$-means on a mixture of 10 Gaussians without overlap. On top of the 500 samples from the mixture there are 150 outliers uniformly distributed in $[-1, 1]^{10}$.

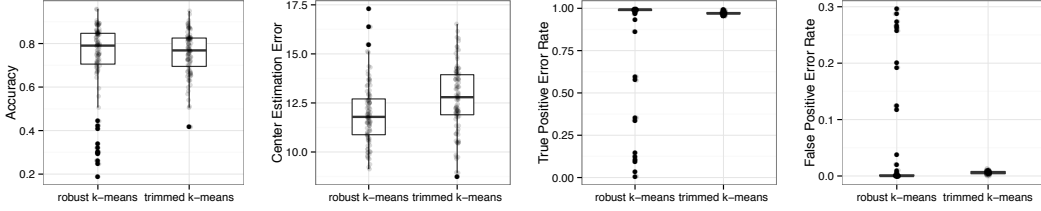

Figure 3: The same setup as in Figure 2 except that the coordinates of each outlier lie in $[-3, 3]^{10}$.

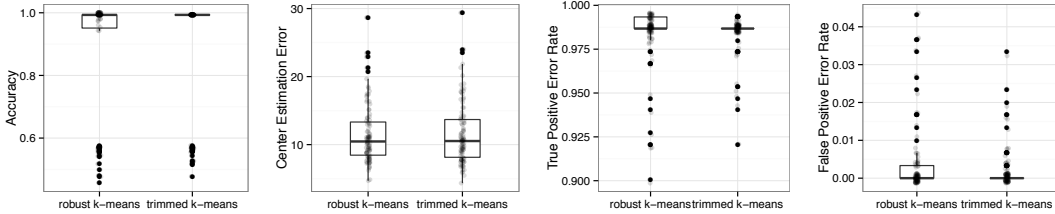

Figure 4: Results on two spherical clusters with equal radius $r$, each one with 150 samples, and centers are at least $4r$ apart. On top of the samples lie 150 outliers uniformly distributed in $[-6, 6]^{10}$.

In Figure 4, we plot the results for the case of two spherical clusters in $\mathbb{R}^{10}$ with equal radius $r$, each one with 150 samples, and centers that are at least $4r$ apart from each other. The inlier samples are in $[-3, 3]^{10}$. The outliers are 150 (half of the dataset is contaminated) and are uniformly distributed in $[-6, 6]^{10}$. The results (accuracy, cluster estimation error, etc) are averages over 150 runs of the experiment. This configuration is a heavy contamination scenario but, due to the structure of the dataset, as expected from Theorem 2, (RKM) performs remarkably well; the same holds for (TKM).

## 6 Conclusions

We provided a theoretical analysis for the robustness and consistency properties of a variation of the classical quadratic $k$-means called robust $k$-means (RKM). As a by-product of the analysis, we derived a detailed description of the optimality conditions for the associated minimization problem. In most cases, (RKM) shares the computational simplicity of quadratic $k$-means, making it a "computationally cheap" candidate for robust nearest neighbor clustering. We show that (RKM) cannot be robust against any type of contamination and any type of datasets, no matter the form of the proximal map we use. If we restrict our attention to "well-structured" datasets, then the algorithm exhibits some desirable noise robustness. As for the consistency properties, we showed that most general results for consistency of quadratic $k$-means still remain valid for this robust variant.

**Acknowledgments**

The author would like to thank Athanasios P. Liavas for useful comments and suggestions that improved the quality of the article.

## Footnotes

[1] For a similar definition for the set of clusters induced by a bounded $\phi$ see also Section 4 in [2].

[2] We call $f$ proper if $f(x) < \infty$ for at least one $x \in \mathbb{R}^n$, and $f(x) > -\infty$ for all $x \in \mathbb{R}^n$; in words, if the domain of $f$ is a nonempty set on which $f$ is finite (see page 5 in [19]).

[3] Accordingly, for a general function $\phi : \mathbb{R} \to [0,\infty)$ to be a Moreau envelope, i.e., $\phi(\cdot) = e_{f_\lambda}(\cdot)$ as defined in (5a) for some function $f_\lambda$, we require that $\phi(\cdot) - \frac{1}{2}|\cdot|^2$ is a concave function (Proposition 1 in [26]).

[4]More than one $\lambda$ can yield the same $r$, but this does not affect our analysis.

[5]See the analysis in [7] about the influence function of (GKM) when $\phi$ is convex.

[6]We use the implementation of trimmed $k$-means in the R package trimcluster [10].

[7]The previous three steps are performed also by another robust variant of $k$-means, the $k$-means$-$ (see [3]).

[8]We use the R toolbox MixSim [14] that guarantees no overlap among the 10 mixtures.

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
