[Supplementary Material · appendix.pdf]

# A  Appendix to: "Robust $k$-means: a theoretical revisit"

## A.1  Proof of Corollary 1

*Proof.* The proof for the first part appears also in the Rejoiner paper of [1] but we repeat it since the second part is a straightforward extension of it. If $s = ||z||_2$, then $e_{F_\lambda}$ in (5b) can be written as

$$e_{F_\lambda}(x) := \min_s \min_{z \in \mathbb{R}^p, ||z||_2 = s} \frac{1}{2}||x - z||_2^2 + f_\lambda(s). \tag{24}$$

The minimizing argument for the inner problem is $z_s = sx/||x||_2$. Substituting this value into (24), we get

$$e_{F_\lambda}(x) = \min_s \frac{1}{2}(||x||_2 - s)^2 + f_\lambda(s). \tag{25}$$

The optimal point in (25) is $s^* = P_{f_\lambda}(||x||_2)$. Recalling that the minimum of $f_\lambda$ is attained at 0 and plugging $s^*$ into $z_s = sx/||x||_2$, we get the proximal map for (5b),

$$\mathsf{P}_{F_\lambda}(x) = \begin{cases} \frac{x}{||x||_2} P_{f_\lambda}(||x||_2), & \text{if } x \neq 0, \\ 0, & \text{otherwise.} \end{cases}$$

With a similar reasoning, we have

$$
\begin{aligned}
e_{F_\lambda}(x) &:= \min_{z \in \mathbb{R}^p} \frac{1}{2}||x - z||_2^2 + F_\lambda(z) \\
&= \min_s \min_{z:||z||_2 = s} \frac{1}{2}||x - z||_2^2 + f_\lambda(s) \quad (\text{since } F_\lambda(\cdot) := f_\lambda(||\cdot||_2)) \\
&= \min_s \frac{1}{2}\left\|x - s\frac{x}{||x||_2}\right\|_2^2 + f_\lambda(s) \\
&= \min_s \frac{1}{2}(||x||_2 - s)^2 + f_\lambda(s) \\
&= e_{f_\lambda}(||x||_2).
\end{aligned}
\tag{26}
$$

$\square$

## A.2  Proof of Lemma 1

We recall some definitions. As already mentioned in the text, in the general non-smooth and non-convex setting, the usual subgradient does not fully characterize the differentiability of (RKM). Instead, we use generalized subgradients. First, we define the regular (or Frechét) subdifferential $\hat{\partial}\Phi(x)$ of a function $\Phi : \mathbb{R}^p \to \mathbb{R}$ at $x$, as the collection of vectors $v$, such that [19]

$$\forall z \in \mathbb{R}^p, \ \Phi(z) \geq \Phi(x) + (z - x)^\top v + o(||z - x||_2).$$

Taking the graphical closure of $\hat{\partial}\Phi(x)$:

$$\partial\Phi(x) := \{v \in \mathbb{R}^p \ : \ \exists x_n \to x, \ \Phi(x_n) \to \Phi(x), \ v_n \in \hat{\partial}\Phi(x_n), \ v_n \to v\},$$

we arrive at the (generalized) subdifferential [19]. The points $\{x : 0 \in \partial\Phi(x)\}$ are the critical points of $\Phi$. Clearly $\hat{\partial}\Phi(x) \subseteq \partial\Phi(x)$ for all $x$. Usually, the sets $\hat{\partial}\Phi(x)$ and $\partial\Phi(x)$ coincide and in this case $\Phi$ is called regular at $x$, but there exist cases where they differ, see also Example 1 in Appendin A.9 (for a thorough presentation of generalized subgradients see Chapter 8 in [19]).

*Proof.* It is easier to transform the initial problem (RKM) into the equivalent form

$$
\min_{\substack{c_1,\ldots,c_k \\ w_1,\ldots,w_n}} \sum_{i=1}^n \sum_{l=1}^k w_i[l] \underbrace{\left\{ \min_{o_i} \frac{1}{2}||x_i - c_l - o_i||_2^2 + f_\lambda(||o_i||_2) \right\}}_{\Phi(x_i - c_l)}
$$

$$\text{subject to } c_l \in \mathbb{R}^p, l = 1, \ldots, k, \tag{SRKM}$$
$$w_i \in \Delta_+^k, i = 1, \ldots, n,$$
$$o_i \in \mathbb{R}^p, i = 1, \ldots, n,$$

where $w_i = (w_i[1], \ldots, w_i[k]) \in \Delta_+^k$, and $\Delta_+^k := \{(t_1, \ldots, t_k) \in \mathbb{R}^k : \sum_{i=1}^k t_i = 1,\ t_i \geq 0 \text{ for all } i\}$. Although in this way we remove the non-smooth term $\min_{1 \leq l \leq k}$ from (RKM), the problem (SRKM) still remains non-smooth due to the presence of $f_\lambda$. The computational cost of introducing $n$ additional $k$-dimensional variables is negligible. If $o_i$ and the set $\{c_1, \ldots, c_k\}$ are fixed to values from the previous iteration, then $w_i$ (at the current iteration) is given by $w_i[l] = 1$, if $||x_i - o_i - c_l||_2 \leq ||x_i - o_i - c_{l'}||_2$ for all $l'$ different from $l$ (see Section 4.1 in [21]).

A quick inspection of (SRKM) reveals that the necessary (generalized) first order optimality condition for the center $c_l$ is

$$0 \in \partial \sum_{i \in A_l} \Phi(x_i - c_l) \subseteq \sum_{i \in A_l} \partial \Phi(x_i - c_l), \text{ for } l = 1, \ldots, k, \tag{27}$$

where $A_l$ is defined in (1) and the inclusion follows from the calculus rules of the generalized subgradients (Theorem 9.13 and Corollary 10.9 in [19]). If $f_\lambda$ is convex, the Moreau envelope $e_{f_\lambda}$ is also convex [19] and, in this case,

$$\nabla \Phi(x) = \nabla e_{F_\lambda}(x) = x - P_{F_\lambda}(x). \tag{28}$$

Since we deal with the general non-convex case, (28) becomes (Theorem 10.13 in [19], Proposition 7 in [26])

$$\partial \Phi(x) \in x - P_{F_\lambda}(x). \tag{29}$$

Expressions (27) and (29) imply that

$$0 \in \sum_{i \in A_l} \partial \Phi(x_i - c_l) \subseteq \sum_{i \in A_l} (c_l - x_i + P_{F_\lambda}(x_i - c_l)), \text{ for } l = 1, \ldots, k. \tag{30}$$

$\square$

## A.3   Proof of Proposition 1

*Proof.* We will show that the optimal solution, when $y$ is sufficiently distant from all the other samples, contains the single-point cluster $\{y\}$; this implies the statement of proposition since $||y||_2$ can grow without bound. Let the solution $\{c_1', \ldots, c_k'\}$ be optimal for $\mathcal{X}_1^n$ under the assumption that $y$ does not form a single-point cluster. We will show that the clustering risk

$$R_n'(c_1', \ldots, c_k') = \sum_{u \in \mathcal{X}_1^n} \min_l \phi(||u - c_l'||_2), \tag{31}$$

with $\phi = e_{f_\lambda}$, is larger than the clustering risk

$$R_n'(\hat{c}_1, \ldots, \hat{c}_k) = \sum_{u \in \mathcal{X}_1^n} \min_l \phi(||u - \hat{c}_l||_2), \tag{32}$$

where $\{\hat{c}_1, \ldots, \hat{c}_k\}$ are defined as

$$\hat{c}_l := \begin{cases} 0, & \text{for } l < k, \\ y, & \text{for } l = k. \end{cases} \tag{33}$$

The set $\{\hat{c}_1, \ldots, \hat{c}_k\}$ defines the clusters $\hat{\mathcal{A}} = \{\hat{A}_1, \ldots, \hat{A}_k\}$ with

$$\hat{A}_1 = \{x_1, \ldots, x_{n-1}\},\ \hat{A}_2 = \{\varnothing\}, \ldots,\ \hat{A}_{k-1} = \{\varnothing\},\ \hat{A}_k = \{y\}. \tag{34}$$

We show that

$$R_n'(\hat{c}_1, \ldots, \hat{c}_k) < R_n'(c_1', \ldots, c_k'). \tag{35}$$

Let $\mathcal{I} = \{x_1, \ldots, x_{n-1}\}$ be any set with $n-1$ points in $\mathbb{R}^p$, which without loss of generality are placed around the origin and can be covered with a ball of radius $\text{rad}(\mathcal{I}) < \lambda$. We augment $\mathcal{I}$ with a point $y$, such that

1. $||y - x_i||_2 \geq K_1, \forall i \in \{1, \ldots, n-1\}$,

for a sufficiently large non-negative constant $K_1$. For $\{\hat{c}_1, \ldots, \hat{c}_k\}$, we have

$$R_n'(\hat{c}_1, \ldots, \hat{c}_k) = \sum_{x_i \in \mathcal{I}} \phi(||x_i||_2) + \phi(||y - y||_2) \tag{36}$$
$$\leq (n-1)\phi(\text{rad}(\mathcal{I})).$$

Now, denote the closest center from $\{c_1', \ldots, c_k'\}$ to a sample $u \in \mathcal{X}_1^n$ by $c'(u)$. Without loss of generality, let $\{y, x_j\} \in A_1'$, for some $x_j \in \mathcal{I}$, and denote $||y - c'(y)||_2 = ||y - c_1'||_2 = \delta$. Then, for the sample $x_j \in A_1'$ and $K_1 \geq 3\text{rad}(\mathcal{I}) + \delta$, we have

$$||x_j - y||_2 \geq K_1, \tag{37}$$

and

$$||x_j - c_1'||_2 = ||x_j - y + y - c_1'||_2$$
$$\geq \Big| ||x_j - y||_2 - ||y - c_1'||_2 \Big| \tag{38}$$
$$\geq K_1 - \delta.$$

For any $x_i \in \mathcal{X}_1^n$, not necessarily in $A_1'$, we have

$$||x_i - c'(x_i)||_2 = ||x_i - x_j + x_j - c'(x_i)||_2$$
$$\geq \Big| ||x_j - c'(x_i)||_2 - ||x_i - x_j||_2 \Big| \tag{39}$$
$$\geq K_1 - \delta - 2\text{rad}(\mathcal{I}),$$

where $x_j \in A_1'$.

Thus, the clustering risk $R_n'$ for $\{c_1', \ldots, c_k'\}$ is at least

$$R_n'(c_1', \ldots, c_k') \geq (n-1)\phi(K_1 - \delta - 2\text{rad}(\mathcal{I})|) + \phi(||y - c_1'||_2)$$
$$\geq (n-1)\phi(\text{rad}(\mathcal{I})) + \phi(||y - c_1'||_2)$$
$$> (n-1)\phi(\text{rad}(\mathcal{I})) \tag{40}$$
$$= R_n'(\hat{c}_1, \ldots, \hat{c}_k),$$

and the claim follows. □

### A.4 Proof of Proposition 2

*Proof.* The proof is closely related to the proof of Theorem 2.9 in [18] with some adaptations. Let the solution $\{c_1', \ldots, c_k'\}$ be optimal for $\mathcal{X}_1^n$, under the condition that $y$ is not discarded as outlier, and recall that $\lambda$ is chosen such that $n \geq r(\lambda) > k + 1$. We will show that the clustering risk

$$R_n'(c_1', \ldots, c_k') = \sum_{u \in \mathcal{X}_1^n} \min_l \phi(||u - c_l'||_2),$$

where $\phi = e_{f_\lambda}$, is larger than the clustering risk

$$R_n'(\hat{c}_1, \ldots, \hat{c}_k) = \sum_{u \in \mathcal{X}_1^n} \min_l \phi(||u - \hat{c}_l||_2)), \tag{41}$$

associated with a solution, say $\{\hat{c}_1, \ldots, \hat{c}_l\}$, which discards $y$ if $y$ is sufficiently distant. Let $d(v, \mathcal{X}^n) = \min_{u \in \mathcal{X}^n} ||v - u||_2$ and $\text{diam}(\mathcal{X}^n) = \max_{v \in \mathcal{X}^n} \min_{u \in \mathcal{X}^n} ||v - u||_2$. Without loss of generality, let $R' = \{x_{i_1}, \ldots, x_{i_{r-1}}, y\}$, $i_j \in \{1, \ldots, n-1\}$ and $j \in \{1, \ldots, r-1\}$ be the set of points that (RKM) reports as inliers when the candidate optimal set of centers is $\{c_1', \ldots, c_k'\}$. Accordingly, let $\hat{R} = \{x_{i_1}, \ldots, x_{i_r}\}$, $i_j \in \{1, \ldots, r\}$ be the set of points that (RKM) reports as inliers when the candidate set of optimal centers is $\{\hat{c}_1, \ldots, \hat{c}_k\}$.

We construct the set of centers $\{\hat{c}_1, \ldots, \hat{c}_k\}$ as

$$\hat{c}_l = \begin{cases} x_{i_r}, & \text{if } d(c_l', \mathcal{X}^n) > \text{diam}(\mathcal{X}^n) \\ c_l', & \text{otherwise.} \end{cases}$$

Next, we show that

$$R'_n(\hat{c}_1, \ldots, \hat{c}_k) < R'_n(c'_1, \ldots, c'_k), \tag{42}$$

if $y$ is such that $d(y, \mathcal{X}^n) > 3\text{diam}(\mathcal{X}^n)$. Compare

$$
\begin{aligned}
R'_n(\hat{c}_1, \ldots, \hat{c}_k) &= \sum_{u \in \mathcal{X}_1^n} \min_l \phi(||u - \hat{c}_l||_2) \\
&= \sum_{u \in \hat{R}} \min_l \phi(||u - \hat{c}_l||_2) + \sum_{u \notin \hat{R}} \min_l \phi(||u - \hat{c}_l||_2) \\
&= \underbrace{\sum_{j=1}^{r} \min_l \phi(||x_{i_j} - \hat{c}_l||_2)}_{r \text{ inliers}} + \underbrace{\sum_{x_i \notin \hat{R}} \min_l \phi(||x_i - \hat{c}_l||_2) + \min_l \phi(||y - \hat{c}_l||_2)}_{n-r \text{ outliers}},
\end{aligned}
\tag{43}
$$

to

$$
\begin{aligned}
R'_n(c'_1, \ldots, c'_k) &= \sum_{u \in \mathcal{X}_1^n} \min_l \phi(||u - c'_l||_2) \\
&= \sum_{u \in R'} \min_l \phi(||u - c'_l||_2) + \sum_{u \notin R'} \min_l \phi(||u - c'_l||_2) \\
&= \underbrace{\sum_{j=1}^{r-1} \min_l \phi(||x_{i_j} - c'_l||_2) + \min_l \phi(||y - c'_l||_2)}_{r \text{ inliers}} + \underbrace{\sum_{x_{i_j} \notin R'} \min_l \phi(||x_{i_j} - c'_l||_2)}_{n-r \text{ outliers}}.
\end{aligned}
\tag{44}
$$

Since $n - r$ samples are outliers, the previous two expressions can be further simplified to

$$R'_n(\hat{c}_1, \ldots, \hat{c}_k) = \sum_{j=1}^{r} \min_l \phi(||x_{i_j} - \hat{c}_l||_2) + (n - r)\phi(\lambda) \tag{45}$$

and

$$R'_n(c'_1, \ldots, c'_k) = \sum_{j=1}^{r-1} \min_l \phi(||x_{i_j} - c'_l||_2) + \min_l \phi(||y - c'_l||_2) + (n - r)\phi(\lambda), \tag{46}$$

and so we compare the quantities

$$\sum_{j=1}^{r} \min_l \phi(||x_{i_j} - \hat{c}_l||_2) \quad \text{and} \quad \sum_{j=1}^{r-1} \min_l \phi(||x_{i_j} - c'_l||_2) + \min_l \phi(||y - c'_l||_2).$$

It turns out that is sufficient to show that

$$||x_{i_j} - x_{i_r}||_2 < ||x_{i_j} - c'_l||_2, \quad \text{for } j < r \text{ when } l \text{ is such that } d(c'_l, \mathcal{X}^n) > \text{diam}(\mathcal{X}^n),$$

(recall that in this case $\hat{c}_l = x_{i_r}$) and

$$||x_{i_r} - c'_l||_2 < ||y - c'_l||_2, \quad \text{otherwise.}$$

If $d(c'_l, \mathcal{X}^n) > \text{diam}(\mathcal{X}^n)$, then

$$||x_{i_j} - x_{i_r}||_2 \leq \text{diam}(\mathcal{X}^n) < d(c'_l, \mathcal{X}^n) \leq ||x_{i_j} - c'_l||_2.$$

If $d(c'_l, \mathcal{X}^n) \leq \text{diam}(\mathcal{X}^n)$, then

$$||y - c'_l||_2 \geq d(y, \mathcal{X}^n) - d(c'_l, \mathcal{X}^n) > 3\text{diam}(\mathcal{X}^n) - d(c'_l, \mathcal{X}^n) \geq \text{diam}(\mathcal{X}^n) + d(c'_l, \mathcal{X}^n) \geq ||x_{i_r} - c'_l||_2.$$

We have shown that the optimal solution rejects the replacement $y$ if $d(y, \mathcal{X}^n) > 3\text{diam}(\mathcal{X}^n)$, implying that the estimated cluster centers depend only on samples from $\mathcal{X}^n$. Observing that every candidate center should lie inside the convex hull of $\mathcal{X}^n$, we deduce that all $c_l$ are bounded. In order to complete the proof we note that, no matter the location of $y$, when $y$ is not sufficiently distant so as to be discarded, i.e., $d(y, \mathcal{X}^n) < 3\text{diam}(\mathcal{X}^n)$, the estimated cluster centers are bounded (since $d(y, \mathcal{X}^n)$ is bounded). $\qquad \square$

## A.5 Proof of Theorem 1

For the proof we need the following technical combinatorial lemma from [18].

**Lemma 2** (Lemma 2.8 in [18]). *Let $k \geq 2$, $d \geq 2$, $q \geq k-2$, and $r = d+k$ be natural numbers and let*

$$M = \{x_1, \ldots, x_d\} \cup \{y_1, y_2\} \cup \{z_1, \ldots, z_q\},$$

*with pairwise disjoint elements $x_i$, $y_h$, and $z_j$. Any partition of a subset of $M$ of size $r$ in $k$ clusters is either of the form*

$$\left\{ \{x_1, \ldots, x_d\}, \{y_1, y_2\}, k-2 \ \text{singletons} \ \{z_j\} \right\}$$

*or has a cluster $C$ that contains some pair $\{x_i, y_h\}$ or some pair $\{z_j, u\}$, $u \neq z_j$.*

*Proof.* The proof follows the lines of the proof of Theorem 2.9 in [18]. We proceed in several steps.

1. Construction of the modification $\mathcal{X}_2^n$. Let $\mathcal{I} = \{x_1, \ldots, x_{r-k}\}$ be any set of $r-k$ points in $\mathbb{R}^p$, placed around the origin, that can be covered with a ball of radius $\text{rad}(\mathcal{I}) < \lambda$, such that $\phi(\text{rad}(\mathcal{I})) \leq \phi(\lambda) - \phi(1)$ (this assumption is needed for the purpose of contradiction). We augment $\mathcal{I}$ with $n - r + k - 2$ samples $\{z_1, \ldots, z_{n-r+k-2}\}$ which we control by a non-negative constant $K_1$ such that $F = \mathcal{I} \cup \{z_1, \ldots, z_{n-r+k-2}\}$ and

   (a) $||x_i - z_j||_2 \geq K_1$ for all $i \in \{1, \ldots, r-k\}$ and all $j \in \{1, \ldots, n-r+k-2\}$;
   (b) $||z_i - z_j||_2 \geq K_1$ for all $i \neq j$.

   The set of $x$'s and $z$'s is of size $n-2$. The dataset $F$ is augmented by two arbitrary points $\{y_1, y_2\}$, which we control by a non-negative constant $K_2$, such that

   (c) $||y_1 - y_2||_2 = 1$;
   (d) $||u - y_h||_2 \geq K_2$, for all $u \in F, h = 1, 2$,

   and thus the modification $\mathcal{X}_2^n = F \cup \{y_1, y_2\}$. We will show that the optimal partition of the dataset $\mathcal{X}_2^n$ into $k$ clusters, when $r \geq k+2$, does not discard the replacements $\{y_1, y_2\}$ if $K_1$ and $K_2$ are sufficiently large.

2. Consider the clustering risk on $\mathcal{X}_2^n$ for the set of centers $\{c_1, \ldots, c_k\}$

$$R'_n(c_1, \ldots, c_k) = \sum_{l=1}^{k} \sum_{u \in A_l} \phi(||u - c_l||_2), \tag{47}$$

   where $\mathcal{A} = \{A_1, \ldots, A_k\}$ denote the corresponding clusters and $u$ is any point from $\mathcal{X}_2^n$. We claim that the clustering risk $R'_n$ is bounded above by a constant that depends only on $\text{rad}(\mathcal{I}), k$, and $r$.

   In order to show this, it is sufficient to find a partition $\hat{\mathcal{A}} = \{\hat{A}_1, \ldots, \hat{A}_k\}$ and vectors $\{\hat{c}_1, \ldots, \hat{c}_k\}$ such that $R'_n(\hat{c}_1, \ldots, \hat{c}_k)$ is bounded from above. Consider the following clusters, $\hat{A}_1 = \mathcal{I}, \hat{A}_2 = \{y_1, y_2\}$, and $\hat{A}_j = \{z_j\}$, for $j = 1, \ldots, k-2$, with centers $\hat{c}_1 = 0, \hat{c}_2 = y_1$, and $\hat{c}_j = z_j$, for $1 \leq j \leq k-2$; note that $\hat{c}_1 = 0$ is the center of the ball covering $\mathcal{I}$. With this configuration, the remaining $n-r$ points (outliers) from $\mathcal{X}_2^n$ are $z_j$'s and, due to (b) above, we have

$$R'_n(\hat{c}_1, \ldots, \hat{c}_k) = R'_n(0, y_1, z_1, \ldots, z_{k-2})$$

$$= \sum_{i \in A_1} \phi(||x_i||_2) + \phi(1) + 0 + (n-r)\phi(\lambda)$$

$$\leq |\mathcal{I}|\phi(\text{rad}(\mathcal{I})) + \phi(1) + (n-r)\phi(\lambda).$$

   The dataset $\mathcal{X}_2^n$, by construction, satisfies the assumptions of Lemma 2 and thus any partition of a set with $r$ in $k$ clusters is one of two kinds:

   $(i)$ $\{\{x_1, \ldots, x_{r-2}\}, \{y_1, y_2\}, k-2 \ \text{singletons} \ \{z_j\}\}$

   or

   $(ii)$ there exists a cluster $A_l$, with $|A_l| \geq 2$, containing some pair $\{x_i, y_h\}$ or some pair $\{z_j, u\}$, $z_j \neq u$.

$$\tag{48}$$

3. We claim that the optimal partition of size $r \geq k + 2$ for $\mathcal{X}_2^n$ is of kind $(i)$. Assume on the contrary that it is of kind $(ii)$ and let $\mathcal{A}' = \{A_1', \ldots, A_k'\}$ and $\{c_1', \ldots, c_k'\}$ be the corresponding set of clusters and centers, respectively. We denote the closest center from $\{c_1', \ldots, c_k'\}$ to a sample $u \in \mathcal{X}_2^n$ by $c'(u)$. Without loss of generality, let $\{x_j, y_h\} \in A_1'$ for some $x_j \in \mathcal{I}, j \in \{1, \ldots, r - k\}$ (the case where $\{y, z_l\} \in A_1'$ is handled analogously). This implies that $||y_h - c'(y_h)||_2 = ||y_h - c_1'||_2 \leq \lambda$, since otherwise $y_h$ would be rejected.

For a point $x_j \in A_1'$ and $K_2 \geq \lambda + 2\text{rad}(\mathcal{I})$, we have

$$K_2 - \lambda \leq ||x_j - c'(x_j)||_2,$$

which follows from

$$\left| ||x_j - y_h||_2 - ||y_h - c'(x_j)||_2 \right| = ||x_j - c'(x_j)||_2,$$

along with assumption 1(d). For any $x_i \in \mathcal{I}$ (not necessarily in $A_1'$),

$$K_2 - \lambda - 2\text{rad}(\mathcal{I}) \leq ||x_i - c'(x_i)||_2,$$

which follows from

$$K_2 - \lambda - 2\text{rad}(\mathcal{I}) = \left| ||x_j - c'(x_i)||_2 - ||x_i - x_j||_2 \right| \leq ||x_i - c'(x_i)||_2,$$

where $x_j \in A_1'$. Thus, due to the contribution of outliers, the clustering risk $R_n'(c_1', \ldots, c_k')$ is at least

$$R_n'(c_1', \ldots, c_k') \geq (r - k)\phi(||K_2 - \lambda| - 2\text{rad}(\mathcal{I})|) + (n - r)\phi(\lambda). \tag{49}$$

For a partition of kind $(i)$, under the previous assumptions, we have

$$R_n'(\hat{c}_1, \ldots, \hat{c}_k) \leq (r - k)\phi(\text{rad}(\mathcal{I})) + \phi(1) + (n - r)\phi(\lambda), \tag{50}$$

and since $\phi(\text{rad}(\mathcal{I})) + \phi(1) < \phi(\lambda)$ we have the upper bound

$$\begin{aligned} R_n'(\hat{c}_1, \ldots, \hat{c}_k) &\leq (r - k)\phi(\text{rad}(\mathcal{I})) + \phi(1) + (n - r)\phi(\lambda) \\ &< (r - k)\phi(\lambda) + (n - r)\phi(\lambda) \\ &= (n - k)\phi(\lambda). \end{aligned} \tag{51}$$

As $K_2 \to \infty$, the clustering risk $R_n'(c_1', \ldots, c_k')$ is at least equal to $(n - r)\phi(\lambda)$ since

$$\begin{aligned} R_n'(c_1', \ldots, c_k') &\geq (r - k)\phi(||K_2 - \lambda| - 2\text{rad}(\mathcal{I})|) + (n - r)\phi(\lambda) \\ &\overset{K_2 \to \infty}{=} (n - k)\phi(\lambda), \end{aligned} \tag{52}$$

and finally, due to (51), we arrived at a contradiction; $R_n'(c_1', \ldots, c_k')$ is greater than $R_n'(\hat{c}_1, \ldots, \hat{c}_k)$. Now, by (d) the difference between $y_h$ and $u \in F$ can be made arbitrarily large and the claim of the theorem follows.

$\square$

## A.6   Proof of Corollary 2

*Proof.* The function $g$ is continuous (by definition), increasing in $[0, \infty)$ (since $\nabla g(x) = x - h(x) \geq 0, \forall x \geq 0$), and unbounded ($\lim_{x \to \infty} \nabla g(x) \to \infty$). The continuity of $e_{f_\lambda}$, along with Lemma 1 and expressions (16)-(17), give a description of the epigraph of $e_{f_\lambda}$ in $[0, \infty)$ in terms of $g$; recall that according to Lemma 1, $\partial e_{f_\lambda}(x) \subseteq x - P_{f_\lambda}(x)$. In $[0, \lambda)$, $e_{f_\lambda}$ shares the same epigraph with $g$ up to an additive constant $b$. At the point $x = \lambda$, the epigraph of $e_{f_\lambda}$ has an inward corner point; at this point, the graph of $g(\cdot) + b$ intersects the graph of the constant function $g(\lambda) + b$; the slope with which $g(\cdot) + b$ intersects with $g(\lambda) + b$ at $x = \lambda$ is $\lambda - P_{f_\lambda}(\lambda) = \lambda - h(\lambda)$. For $x > \lambda$, $e_{f_\lambda}$ is constant and equal to $g(\lambda) + b$. Finally, we note that the slope with which the graph of the constant function $g(\lambda) + b$ intersects the graph of $g(\cdot) + b$ at $x = \lambda$ is $\lambda - P_{f_\lambda}(\lambda) = \lambda - \lambda = 0$ since $g(\lambda) + b$ is constant. Hence, the function $e_{f_\lambda}(x)$ is expressed as

$$e_{f_\lambda}(x) = \min\{g(|x|), g(\lambda)\},$$

which, due to the monotonicity of $g$, is equivalent to $g(\min\{|x|, \lambda\})$. $\square$

## A.7 Proof of Theorem 2

*Proof.* Let $\mathcal{A}'(\mathcal{X}^n) = \{A'_1, \ldots, A'_k\}$ be the output clustering of robust $k$-means with centers $\{c'_1, \ldots, c'_k\}$ that minimize $R'_n$, i.e., $c'_l = \arg\min_{c \in \mathbb{R}^p} \sum_{x \in A'_l} e_{f_\lambda}(||x-c||_2)$, and $R'_n(c'_1, \ldots, c'_k) \leq R'_n(c''_1, \ldots, c''_k)$ for all $c''_l \neq c'_l$. Denote as $c'(x)$ the center from $\{c'_1, \ldots, c'_k\}$ to which $x$ is closest. For $l \leq k$, let $b_l$ represent the center of the ball $B_l$ and $D_l$ represent a ball of radius $\frac{v}{2} - r$ centered at $b_l$, i.e., $B_l = B(b_l, r)$ and $D_l = B(b_l, \frac{v}{2} - r)$. Note that, the assumption on $v$ implies $\frac{v}{2} - r > r$.

The following proof is given by means of a contradiction and closely follows Lemma 2 in [2]. We define the families of sets $\mathcal{D}_1, \mathcal{D}_2,$ and $\mathcal{D}_3$ as

- $\mathcal{D}_1 = \{D_l | D_l \text{ does not contain any } c'_j\}$,

- $\mathcal{D}_2 = \{D_l | D_l \text{ contains more that one } c'_j\}$, and,

- $\mathcal{D}_3 = \{D_l | D_l \text{ contains exactly one } c'_j\}$.

Since $D_l$ are pairwise disjoint, $|\mathcal{D}_1| \geq |\mathcal{D}_2|$. We claim that every $D_l$ covers exactly one center $c'_j, j \in \{1, \ldots, k\}$. Assume in search for a contradiction that $\mathcal{D}_1 \neq \emptyset$. Consider all $x \in D_l$ (and $x \in \mathcal{I}$) with $D_l \in \mathcal{D}_1$; then, we have $||x - c'(x)||_2 > \frac{v}{2} - 2r$. To see this, note that if $D_l \in \mathcal{D}_1$, then

$$||c'_l - b_l||_2 > \frac{v}{2} - r \overset{x \in B_l}{\Longleftrightarrow} ||c'_l - b_l - x + x||_2 > \frac{v}{2} - r \Longleftrightarrow$$

$$||c'_l - x||_2 + ||b_l - x||_2 > \frac{v}{2} - r \Longleftrightarrow ||c'_l - x||_2 > \frac{v}{2} - 2r,$$

where the last inequality follows from $||x - b_l||_2 \leq r$ when $x \in B_l$.

Consider the following set of centers: $\{c''_1, \ldots, c''_k\}$. If $D_l \in \mathcal{D}_3$ and $c'_j \in D_l$, then we set $c''_l = c'_j$, else we set $c''_l = b_l$ (this is the case where $D_l \in \mathcal{D}_1$ or $D_l \in \mathcal{D}_2$). For this set of centers, we have

$$R'_n(c''_1, \ldots, c''_k) = \sum_{x \in \mathcal{I}} e_{f_\lambda}(||x - c''(x)||_2) + \sum_{x \in \mathcal{X}^n \setminus \mathcal{I}} e_{f_\lambda}(||x - c''(x)||_2)$$

$$= \sum_{D_l \in \mathcal{D}_3} \sum_{x \in B_l} e_{f_\lambda}(||x - c'(x)||_2) + \sum_{D_l \in \mathcal{D}_2} \sum_{x \in B_l} e_{f_\lambda}(||x - b_l||_2)$$

$$+ \sum_{D_l \in \mathcal{D}_1} \sum_{x \in B_l} e_{f_\lambda}(||x - b_l||_2) + \sum_{x \in \mathcal{X}^n \setminus \mathcal{I}} e_{f_\lambda}(||x - c''(x)||_2)$$

$$\leq R'_n(c'_1, \ldots, c'_k) + \sum_{D_l \in \mathcal{D}_2} \sum_{x \in B_l} (e_{f_\lambda}(||x - b_l||_2) - e_{f_\lambda}(||x - c'(x)||_2))$$

$$+ \sum_{D_l \in \mathcal{D}_1} \sum_{x \in B_l} (e_{f_\lambda}(||x - b_l||_2) - e_{f_\lambda}(||x - c'(x)||_2))$$

$$+ \sum_{x \in \mathcal{X}^n \setminus \mathcal{I}} e_{f_\lambda}(||x - c''(x)||_2)$$

$$\leq R'_n(c'_1, \ldots, c'_k) + \sum_{D_l \in \mathcal{D}_1} |\mathcal{I}|(e_{f_\lambda}(||x - b_l||_2) - e_{f_\lambda}(||x - c'(x)||_2))$$

$$+ \sum_{D_l \in \mathcal{D}_2} |\mathcal{I}| e_{f_\lambda}(||x - b_l||_2) \quad + \sum_{x \in \mathcal{X}^n \setminus \mathcal{I}} e_{f_\lambda}(||x - c''(x)||_2)$$

$$\leq R'_n(c'_1, \ldots, c'_k) + \rho_1 |\mathcal{D}_1||\mathcal{I}| \underbrace{(e_{f_\lambda}(||x - b_l||_2) - e_{f_\lambda}(||x - c'(x)||_2))}_{<0}$$

$$+ \rho_2 |\mathcal{D}_2||\mathcal{I}| e_{f_\lambda}(||x - b_l||_2) + \sum_{x \in \mathcal{X}^n \setminus \mathcal{I}} e_{f_\lambda}(||x - c''(x)||_2)$$

$$\leq R'_n(c'_1, \ldots, c'_k) + |\mathcal{D}_1||\mathcal{I}| \left( (\rho_1 + \rho_2) g(r) - \rho_1 g(\frac{v}{2} - 2r) \right)$$

$$+ |\mathcal{X}^n \setminus \mathcal{I}| g(\lambda) \quad (\text{since } |\mathcal{D}_1| \geq |\mathcal{D}_2|)$$

$$< R'_n(c'_1, \ldots, c'_k) \quad (\text{by the choice of } \lambda).$$

This forms a contradiction since we assumed that $\{c'_1, \ldots, c'_k\}$ are optimal. Hence, the set $\mathcal{D}_1$ is empty, which implies that $\mathcal{D}_2$ is empty, and every $D_l$ covers exactly one center $c'_l$. For all $l \neq j$ and for all $x \in B_l$ we have $||x - c'_l||_2 \leq \frac{v}{2} \leq ||x - c'_j||_2$ and therefore the output of (RKM) when restricted on the inlier samples is $\mathcal{A}'(\mathcal{X}^n)|\mathcal{I} = \{B_1, \ldots, B_k\} \cup \{\emptyset\}$. The proof is now finished. $\square$

## A.8 Proof of Theorem 3

The complete proof of the theorem is a typical consistency proof for an estimator defined by optimization of a random criterion function. First, one forces the optimal solution, in our case $\mathcal{C}^*$, to lie into a restricted (often compact) region. That is usually the hardest part of the proof. Otherwise, one should assume that $\mu$ is supported on a closed ball of radius $M$ centered at the origin (this assumption is also known as the 'peak power' constraint). Fortunately, the hardest part has been done; Pollard [15] shows that when $\phi$ is a general, lower semicontinuous, and non-decreasing function, like the Moreau envelopes we work with, the optimal population set of centers $\mathcal{C}^*$ lies inside a compact ball $B(M) \subset \mathbb{R}^p$ with radius $M > 0$ centered at the origin (it turns out that the empirical optimal set $\hat{\mathcal{C}}$ lies also in $B(M)$). Then, an appeal to a uniform strong law over the restricted region gives the desired results. In what follows, assuming that $\hat{\mathcal{C}}$ and $\mathcal{C}^*$ lie in $B(M)$ and are unique, we make slight modifications to the non-asymptotic consistency result in ([11], Theorem 3) and show that it holds also for bounded $\phi$ like the Moreau envelopes associated with unbiased proximal maps.

*Proof.* The main probabilistic tool for proving the loss consistency of robust $k$-means, i.e., $\lim_{n \to \infty} R'(\hat{\mathcal{C}}) \to R'(\mathcal{C}^*)$, is the statement of uniform convergence of the empirical to the true measure. This can be easily seen by the so called VC (Vapnik-Chervonenkis) inequality (Lemma 8.2 in [4]),

$$
\begin{aligned}
R'(\hat{\mathcal{C}}) - R'(\mathcal{C}^*) &= R'(\hat{\mathcal{C}}) - \inf_{c_1, \ldots, c_k} R'(c_1, \ldots, c_k) \\
&= R'(\hat{\mathcal{C}}) - R'_n(\hat{\mathcal{C}}) + R'_n(\hat{\mathcal{C}}) - \inf_{c_1, \ldots, c_k} R'(c_1, \ldots, c_k) \\
&\leq R'(\hat{\mathcal{C}}) - R'_n(\hat{\mathcal{C}}) + \sup_{c_1, \ldots, c_k} |R'_n(c_1, \ldots, c_k) - R'(c_1, \ldots, c_k)| \\
&\leq 2 \sup_{c_1, \ldots, c_k} |R'_n(c_1, \ldots, c_k) - R'(c_1, \ldots, c_k)|.
\end{aligned}
\tag{53}
$$

For the passage to the first inequality note that,

$$
\begin{aligned}
R'_n(\hat{\mathcal{C}}) - \inf_{c_1, \ldots, c_k} R'(c_1, \ldots, c_k) &= R'_n(\hat{\mathcal{C}}) + \sup_{c_1, \ldots, c_k} -R'(c_1, \ldots, c_k) \\
&= \sup_{c_1, \ldots, c_k} \left\{ \inf_{b_1, \ldots, b_k} R'_n(b_1, \ldots, b_k) - R'(c_1, \ldots, c_k) \right\} \\
&\leq \sup_{c_1, \ldots, c_k} \left\{ R'_n(c_1, \ldots, c_k) - R'(c_1, \ldots, c_k) \right\}.
\end{aligned}
$$

Therefore the empirical risk is (on any bounded set of cluster centers) a continuous function of the cluster centers. Thus, an upper bound for $\sup_{c_1, \ldots, c_k} |R'_n - R'|$ provides a bound for $R'(\hat{\mathcal{C}}) - R'(\mathcal{C}^*)$.

For every set of $k$ centers $\mathcal{C} = \{c_1, \ldots, c_k\}$ with $c_l \in B(M) \subset \mathbb{R}^p$, we define the function

$$
\phi_{\mathcal{C}}(x) := \min_{1 \leq l \leq k} \phi(||x - c_l||_2).
\tag{54}
$$

Note that $0 \leq \phi_{\mathcal{C}}(x) \leq \phi(\lambda)$, since $\phi$ is upper-bounded (by construction) for all $x \in B(M)$ and $c_l \in B(M)$. Exploiting the identity $\mathbb{E}Y = \int_0^\infty \mathbb{P}\{Y \geq t\} dt$, for any non-negative real random

variable $Y$, we have (see also Lemma 29.1 in [4]))

$$\sup_{c_1,\ldots,c_k} |R'_n(c_1,\ldots,c_k) - R'(c_1,\ldots,c_k)| = \sup_{c_1,\ldots,c_k} \left| \frac{1}{n} \sum_{i=1}^{n} \phi_{\mathcal{C}}(x_i) - \mathbb{E}\phi_{\mathcal{C}}(x) \right|$$

$$= \sup_{c_1,\ldots,c_k} \left| \int_0^{\phi(\lambda)} \left( \frac{1}{n} \sum_{i=1}^{n} \mathbb{1}_{\{\phi_{\mathcal{C}}(x_i)>t\}} - \mathbb{P}\{\phi_{\mathcal{C}}(x) > t\} \right) dt \right|$$

$$\leq \phi(\lambda) \sup_{\substack{c_1,\ldots,c_k \\ t>0}} \left| \frac{1}{n} \sum_{i=1}^{n} \mathbb{1}_{\{\phi_{\mathcal{C}}(x_i)>t\}} - \mathbb{P}\{\phi_{\mathcal{C}}(x) > t\} \right|$$

$$\leq \phi(\lambda) \sup_{A \in \mathcal{F}_k} |\mu_n(A) - \mu(A)|,$$

$$(55)$$

where $A$ varies over the family of sets $\mathcal{F}_k$ in $\mathbb{R}^p$ defined as

$$\mathcal{F}_k := \{\{x : \phi_{\mathcal{C}}(x) > t\}, \text{ where } \mathcal{C} = \{c_1,\ldots,c_k\}, \ c_l \in B(M), \text{ and } t > 0\},$$

and $\mu_n(A) = \frac{1}{n}\sum_{i=1}^{n} \mathbb{1}_{\{x_i \in A\}}$ is the empirical measure associated with $\mathcal{X}^n = \{x_1,\ldots,x_n\}$. Next, we recall the concept of *shatter coefficient* and *VC dimension* for a collection of measurable sets.

**Definition 3** (shatter coefficient [4]). *Let $\mathcal{A}$ be a family of sets. For $\{x_1,\ldots,x_n\} \subset \mathbb{R}^p$, let $N_{\mathcal{A}}(x_1,\ldots,x_n)$ be the number of different sets in*

$$\{\{x_1,\ldots,x_n\} \cap A; \ A \in \mathcal{A}\}.$$

*The $n$-th shatter coefficient of $\mathcal{A}$ is*

$$s(\mathcal{A},n) = \max_{x_1,\ldots,x_n} N_{\mathcal{A}}(x_1,\ldots,x_n).$$

*That is, the shatter coefficient is the maximal number of different subsets of $n$ points that can be picked out by sets belonging in $\mathcal{A}$.*

**Definition 4** (VC dimension [4]). *Let $\mathcal{A}$ be a collection of sets with $|\mathcal{A}| \geq 2$. The largest integer $k \geq 1$ for which $s(\mathcal{A},k) = 2^k$ is denoted by $V_{\mathcal{A}}$, and is called the VC dimension of the class $\mathcal{A}$.*

Regarding inequality (55), Vapnik-Chervonenkis's Theorem 12.5 in [4] states that

$$\mathbb{P}\left\{ \sup_{c_1,\ldots,c_k} |R'_n(c_1,\ldots,c_k) - R'(c_1,\ldots,c_k)| > \epsilon \right\} \leq \mathbb{P}\left\{ \sup_{A \in \mathcal{G}_k} |\mu_n(A) - \mu(A)| > \frac{\epsilon}{\phi(\lambda)} \right\} \tag{56}$$

$$\leq 8s(\mathcal{F}_k,n)e^{-\frac{n\epsilon^2}{32\phi(\lambda)^2}}.$$

The previous bound is meaningful only when $V_{\mathcal{F}_k}$ is finite and does not increase too fast with $n$ (see also the last part of Theorem 4.3 in [11]). For example, if $s(A,n)$ is a polynomial of degree $a$ with respect to $n$ i.e., $s(A,n) = n^a$, which we show to be the case next, then $\lim_{n\to\infty} n^a e^{-n} \to 0$.

The set $\mathcal{F}_k$ is described by the sets with points $x$ for which

$$\phi(||x - c_l||_2) > t, \text{ for all } \quad l = 1,\ldots,k, \tag{57}$$

and thus each $A \in \mathcal{F}_k$ is the intersection of $k$ sets of the form $\{x : \phi(||x-c_l||_2) > t\}$ for $l = 1,\ldots,k$. Let $\mathcal{F}$ denote the collection of all the latter sets, i.e.,

$$\mathcal{F} := \{x : \phi(||x - c||_2) > t\}, \text{ for some } c \in \mathbb{R}^p. \tag{58}$$

From Theorem 13.5 in [4], we know that the $n$-th shatter coefficient of $\mathcal{F}_k$ is upper bounded by

$$s(\mathcal{F}_k,n) \leq s(\mathcal{F},n)^k \text{ for all } n,k \geq 1. \tag{59}$$

So, what remains is to find the shatter coefficient of the sets in (58).

Consider the set of points $\{(x,t) \in \mathbb{R}^{d+1} : \phi(||x - c||_2) \geq t\}$. Due to the monotonicity of $\phi(||\cdot -c||_2)$, a point $(x_0,t_0) \in \mathbb{R}^{p+1}$ with $t_0 \geq 0$ belongs to the latter set if and only if

$$||x_0 - c||_2 \geq a(t_0), \tag{60}$$

where $a(t_0)$ denotes the smallest value of $a$ for which $\phi(a) \geq t_0$. In view of the previous inequality, we conclude that, from a collection of points $\{(x_i, t_i)\}_{i=1}^n$ in $\mathbb{R}^{p+1}$, only those points satisfying

$$||x_i||_2^2 - 2x_i^\top c + ||c||_2^2 - a(t_i)^2 \geq 0,$$

belong to $\{(x, t) \in \mathbb{R}^{d+1} : \phi(||x - c||_2) \geq t\}$. Construct from $\{(x_i, t_i)\}_{i=1}^n$ the points $z_i = (x_i, ||x_i||_2^2 - a(t_i)^2)$. On $\mathbb{R}^{p+1}$ define a vector space $\mathcal{G}$ of functions

$$\mathcal{G} := \left\{ g : \mathbb{R}^{p+1} \to \mathbb{R} \text{ such that } g_{\beta,\gamma,\delta}(x, t) = \beta x + \gamma t + \delta \right\}, \tag{61}$$

with parameters $\beta \in \mathbb{R}^p$ and $\gamma, \delta \in \mathbb{R}$. From Theorem 13.9 in [4], the sets

$$\{(x, t) : g_{\beta,\gamma,\delta}(x, t) \geq 0\}$$

pick out only a polynomial number of subsets from $\{z_i\}_{i=1}^n$ and more specially $V_{\mathcal{G}} \leq p + 2$. Those sets corresponding to functions in $\mathcal{G}$ with $\beta = -2c$, $\gamma = 1$, and $\delta = ||c||_2^2$ pick out even fewer subsets from $\{z_i\}_{i=1}^n$; fortunately, we arrive at the bound,

$$V_{\mathcal{F}} < V_{\mathcal{G}} \leq p + 2. \tag{62}$$

In view of (62) and Theorem 13.3 in [4], an upper bound for the shatter coefficient of the class $\mathcal{F}_k$ in (59) is

$$s(\mathcal{F}_k, n) < \left( \frac{ne}{p+2} \right)^{k(p+2)}. \tag{63}$$

Plugging the previous bound in (56), we get

$$\mathbb{P} \left\{ \sup_{c_1, \ldots, c_k} |R'_n(c_1, \ldots, c_k) - R'(c_1, \ldots, c_k)| > \epsilon \right\} < 8 \left( \frac{ne}{p+2} \right)^{k(p+2)} e^{-\frac{n\epsilon^2}{32\phi(\lambda)^2}}, \tag{64}$$

which gives

$$\mathbb{P} \left\{ R'_n(\hat{\mathcal{C}}) - R'(\mathcal{C}^*) > \epsilon \right\} < 8 \left( \frac{ne}{p+2} \right)^{k(p+2)} e^{-\frac{n\epsilon^2}{128\phi(\lambda)^2}}. \tag{65}$$

We conclude almost sure convergence of $R'_n$ to $R'$ by noting that

$$\sum_{n=0}^{\infty} \mathbb{P}\{R'_n(\hat{\mathcal{C}}) - R'(\mathcal{C}^*) > \epsilon\} < \infty,$$

for all $\epsilon > 0$ and thus $R'(\hat{\mathcal{C}}) \overset{n\to\infty}{\longrightarrow} R'(\mathcal{C}^*)$ a.s.

Now, we sketch the proof for strong consistency of the cluster centers. From Proposition 5.2.1 in [23], we know that $\hat{\mathcal{C}} \overset{P}{\to} \mathcal{C}^*$ if $i)$ $\sup_{\mathcal{C}} |R'_n(\mathcal{C}) - R'(\mathcal{C})| \overset{P}{\to} 0$ and $ii)$ the map $\mathcal{C} \mapsto R'(\mathcal{C})$ is continuous on $\mathbb{R}^p$. [9] So what remains to conclude is the continuity of the map $R'$ which sends a set of centers $\mathcal{C}$ to the values of the clustering risk $R'(\mathcal{C})$. The continuity of $R'$, with respect to Hausdorff metric, is also proved in [15]. □

## A.9 Discussion of coordinate descent for (RKM)

Finding the true optimal partition for a fixed number of sets in a given $p$-dimensional space is known to be an NP-hard problem [8] and so we rely on approximation algorithms to find a locally optimal solution for (RKM). As a result, all current algorithms for robust $k$-means [5, 24] are nothing else but the coordinate descent method or else alternating optimization [22]. Applying coordinate optimization on (SRKM) (see proof of Lemma 1), at each iteration we:

1. Fix $w_i, c_l$ and minimize with respect to $o_i$ i.e., set $o_i = P_{F_\lambda}(x_i - c_l)$ when $x_i \in A_l$.

2. Fix $o_i$ and minimize with respect to $w_i, c_l$ i.e., we solve a $k$-means problem with $x_i$ replaced by $x_i - P_{F_\lambda}(x_i - c_l)$.

3. Repeat the previous two steps until convergence of the centers.

Figure 5: The graph of $\frac{1}{5}\sum_{x_i \in \mathcal{X}} \Phi(x_i - c)$ for the point set $\mathcal{X} = \{-2, -1, 1, 2, 4\}$ (red circles) in Example 1 when $\Phi = e_{\lambda|\cdot|_0}$. The $x$-axis represents the variable $c$ while, the $y$-axis represents the values of the objective function $\sum_{x_i \in \mathcal{X}} e_{\lambda|\cdot|_0}(x_i - c)$ as a function of $c$. The subgradient set at $c = 0$ is empty (left cross mark) while at $c = 0.8$ (right cross mark), it contains only the zero vector. The two dashed line arrows constitute the non-convex tangent cone of the epigraph at $0$, i.e., the tangent cone consists of only those two elements.

If the computational complexity of the proximal map is low, then (RKM) retains the computational simplicity of the classical quadratic $k$-means algorithm [12].

Recently, a lot of work has been done in understanding the convergence properties of coordinate descent methods for the case of Lipschitz functions $\Phi : \mathbb{R}^p \to \mathbb{R}$, see [25] and references therein. *Does all proximal maps $P_{f_\lambda}$ guarantee the convergence of coordinate descent algorithms to a local minimum or even a stationary point?* The answer to the previous question is *no* and recalling one of the basic assumptions on convergence of alternating optimization, i.e. every coordinate minimum should be unique [22], it is not a surprise that the causes for this instability are multivaluedness of $P_{f_\lambda}$ along with possible discontinuities.

Roughly speaking, two main assumptions must be met for convergence of coordinate descent to a *regular* stationary point, i.e., a point $x$ where the regular directional derivative $\Phi'(x; d) := \liminf_{t \downarrow 0} (\Phi(x + td) - \Phi(x))/t$ is non-negative. Either $\Phi$ should be regular at every point $x$ in its domain, i.e., $\partial\Phi(x) = \hat{\partial}\Phi(x)$, for all $x$ in the domain of $\Phi$, or $\Phi$ should have a unique minimum along every coordinate [22]. For (RKM) and the class of proximal maps in (16), neither of the previous two assumptions is met and so the price we have to pay for using coordinate descent is the possible convergence to a non-stationary point, i.e., a point where the directional derivative as previously defined is negative; see Example 1 below.

**Example 1.** *Consider the non-convex $l_0$ (pseudo) norm, $\lambda|x|_0 = \frac{\lambda^2}{2}\mathbf{1}_{x \neq 0}$ and the associated hard thresholding proximal map (12). Note that $P_{\lambda|\cdot|_0}(x)$ is discontinuous and multivalued at $|x| = \lambda$ implying the existence of two minimizing arguments for (5b). Resorting to the definition, we see that the Moreau envelope of the $l_0$-norm is given by*

$$e_{\lambda|\cdot|_0}(x) = \frac{1}{2}\min\{x^2, \lambda^2\}.$$

*Next, consider the set of points $\mathcal{X} = \{-1, -2, 1, 2, 4\}$ and the problem,*

$$\min_c \sum_{x_i \in \mathcal{X}} \frac{1}{5}\Phi(x_i - c),$$

*where $\Phi = e_{\lambda|\cdot|_0}$ and $\lambda = 4$. Coordinate descent for the previous problem takes the form of the following update rule,*

$$c^{t+1} \leftarrow c^t - \frac{1}{5} \sum_{x_i \in \mathcal{X}} (x_i - P_{\lambda|\cdot|_0}(x_i - c^t)).$$

*A common initialization method for k-means is sampling from data, i.e., k random points are sampled uniformly from the dataset as initial centers. When the initial point is $c^0 \in \{-2, -1\} < 0$, the previous update scheme converges to $c = 0$ and the objective function equals,*

$$\sum_{x_i \in \mathcal{X}} \frac{1}{5} \Phi(x_i - 0) = \sum_{x_i \in \mathcal{X}} \frac{1}{5} e_{\lambda|\cdot|_0}(x_i - 0) = 2.6.$$

*When $c^0 \in \{1, 2, 4\} \geq 0$ the solution is $c = 0.8$ and,*

$$\sum_{x_i \in \mathcal{X}} \frac{1}{5} \Phi(x_i - 0.8) = \sum_{x_i \in \mathcal{X}} \frac{1}{5} e_{\lambda|\cdot|_0}(x_i - 0.8) = 2.28,$$

*which is the global minimum value; Figure 5 shows the graph of $\sum_{x_i \in \mathcal{X}} \frac{1}{5} e_{\lambda|\cdot|_0}(x_i - c)$. Although for $c = 0$, both sets $\sum_{x_i \in \mathcal{X}} (x_i - P_{\lambda|\cdot|_0}(x_i))$ and $\sum_{x_i \in \mathcal{X}} \partial e_{\lambda|\cdot|_0}(x_i)$ include the 0 element (horizontal dashed arrow in Figure 5), $c = 0$ is not a local minimum. In fact, it is a saddle point in the general sense, i.e., $0 \in \sum_{x_i \in \mathcal{X}} \partial e_{\lambda|\cdot|_0}(x_i - 0)$, but not in the regular sense, i.e., $0 \notin \sum_{x_i \in \mathcal{X}} \hat{\partial} e_{\lambda|\cdot|_0}(x_i - 0)$ since, at that point the regular subgradient set is empty, $\sum_{x_i \in \mathcal{X}} \hat{\partial} e_{\lambda|\cdot|_0}(x_i - 0) = \emptyset$. It useful to recall the definition of the generalized subgradient for a function g at a point x as the set of all vectors $u \in \partial g(x)$ such that $u^\top s \leq 0$, $s \in T(x)$, where $T(x)$ is the general tangent cone of the epigraph of g at x; in this example, the general tangent cone of $\sum_{x_i \in \mathcal{X}} e_{\lambda|\cdot|_0}(x_i - c)$ at $c = 0$ is non-convex [19] (see explanation in Figure 5).*

In Example 1, the epigraph of $e_{f_\lambda}$ at $c = 0$ has an inward corner (Figure 5) indicating absence of regularity at that point (with the notation of [19], the regular normal cone $\hat{N}(c)$ and the generalized normal cone $N(c)$ of the epigraph of $\sum_{x_i \in \mathcal{X}} \frac{1}{5} e_{\lambda|\cdot|_0}(x_i - c)$ at $c = 0$ are different). Next, we repeat the previous example with a continuous, single valued, and monotone proximal map. An immediate corollary of single-valuedness and continuity is convergence of coordinate descent to a stationary center $c$, i.e., $0 \in \sum_{x_i \in \mathcal{X}} \frac{1}{5} \nabla e_{\lambda|\cdot|_0}(x_i - c)$, since in this case the set of regular and general subgradients coincides.

**Example 2.** *Tukey's non-convex penalty function [1, 20] comes from the literature of robust statistics and has proximal map*

$$P_{f_\lambda}(x) = \begin{cases} x - x(1 - (\frac{x}{\lambda})^2)^2, & \text{if } |x| \leq \lambda, \\ x, & \text{otherwise.} \end{cases}$$

*An explicit representation for $f_\lambda$ for the previous proximal map can be derived following the three-step construction described in Section 3 in [20]. The Moreau envelope of Tukey's penalty is*

$$e_{f_\lambda}(x) = \begin{cases} \frac{\lambda^2}{6}(1 - (1 - (\frac{x}{\lambda})^2)^3), & \text{if } |x| \leq \lambda, \\ \frac{\lambda^2}{6}, & \text{otherwise,} \end{cases}$$

*and is known as the biweight function. Since $P_{f_\lambda}$ is continuous and single-valued, every vector $P_{f_\lambda}(x_i - c)$ is associated with a gradient vector $\nabla \Phi(x_i - c)$ and, therefore, we expect convergence of coordinate descent to a stationary point in the regular sense (since we deal with the smooth case, every generalized stationary point is also a regular stationary point). Consider the same set of points $\mathcal{X}$ as in Example 1, $k = 1$, $\lambda = 4$ and the same initialization method i.e., sampling from the dataset. Iterating a scheme of the form*

$$c_1^{t+1} \leftarrow c^t - \frac{1}{5} \sum_{x_i \in \mathcal{X}} (x_i - P_{f_\lambda}(x_i - c^t)),$$

*until there is no change in $c^t$, no matter the initial point, the algorithm converges to $c = 0$ that happens to be the unique global minimum of*

$$\sum_{x_i \in \mathcal{X}} \frac{1}{5} e_{f_\lambda}(x_i - c),$$

*see also Figure 6.*

Figure 6: The graph of $\frac{1}{5}\sum_{x_i \in \mathcal{X}} e_{f_\lambda}(x_i - c)$ when $f_\lambda$ is Tukey's penalty for the point set $\{-2, -1, 1, 2, 4\}$ (red circles) in Example 2.

## Footnotes

[9]The symbol $\overset{P}{\to}$ denotes convergence in probability.