[Reviews · NeurIPS 2016]

Reviewer 1

Summary

In this paper the author studied theoretic properties of the robust k-means (RKM) formulation proposed in [5,23]. They first studied the robustness property, showing that if the f_\lambda function is convex, the one outlier is sufficient to break down the algorithm; and if f_\lambda need not be convex, then two outliers can breakdown the algorithm. On the other hand, under some structural assumptions on the non-outliers, then a non-trivial breakdown point can be established for RKM. The authors then study the consistency issue, generalising consistency results that are known for convex f_lambda to non convex f_\lambda.

Qualitative Assessment

In this paper the author studied theoretic properties of the robust k-means (RKM) formulation proposed in [5,23]. They first studied the robustness property, showing that if the f_\lambda function is convex, the one outlier is sufficient to break down the algorithm; and if f_\lambda need not be convex, then two outliers can breakdown the algorithm. On the other hand, under some structural assumptions on the non-outliers, then a non-trivial breakdown point can be established for RKM. The authors then study the consistency issue, generalising consistency results that are known for convex f_lambda to non convex f_\lambda. My main concern of the paper is that the results appear very specific and I am not entirely sure whether they will appeal to a more general audience in machine learning. The results are about the RKM formulation, a formulation IMO has not established itself as a standard method. Also, it does not appear that the technique to develop these results may easily adapt to other methods either. I have a question about section 3.3. From its current exposition, the main result of the subsection appears to be re-capping what is known in literature. Indeed there is no formal theorem presented in section 3.3, and the last paragraph which looks like the main result ends with “see also Theorem 3, section 5 in [2]”. Regarding the consistency results, in the abstract the authors mention that they extend non-asymptotic results to the non-convex f_\lambda case. However, the main theorem of this paper - Theorem 2 - reads like an asymptotic convergence result. This might be my misunderstanding, and I would appreciate any clarification from the authors. About the simulation, I am not sure what conclusion can be drawn, in particular about comparison with trimmed k-means.

Confidence in this Review

2-Confident (read it all; understood it all reasonably well)


Reviewer 2

Summary

The authors study robustness and consistency of the Robust k-means (RKM) objective. Using the framework of the universal breakdown point, the authors show that RKM is not robust! To be precise, the universal breakdown point of RKM for unbiased proximal map is 2/n, and for the biased one it is 1/n. In other words, even 2 outliers are enough to breakdown some centers (that is, move them arbitrarily far). However, they contrast this by showing that RKM is robust on well-clustered data, if we use the notion of (\rho_1, \rho_2)-balanced data sets by Ben-David and Haghtalab (ICML'14). The authors also show that the consistency property of the usual k-means remains valid even for the more general RKM.

Qualitative Assessment

The paper is well-written, the problem is well-motivated. The results are not spectacular but should still be of interest to part of the community working on clustering. The experiments are good but I did not pay a great attention to them as the main contribution of this paper is theoretical. Study of robust variants of clustering problems (and k-means in particular) that incorporate outliers has potential for big impact. I would recommend acceptance. Minor typos etc. Line 23 -- when "measured" according

Confidence in this Review

3-Expert (read the paper in detail, know the area, quite certain of my opinion)


Reviewer 3

Summary

The authors analyzed the robustness and consistency of the algorithm robust k-means. They showed that two outliers are enough to breakdown this algorithm. However, if the data are well-structured, it remains robust to outliers. Furthermore, they proved that even when robust k-means involves a non-convex penalty function, its consistency property remains valid.

Qualitative Assessment

The problems studied in this manuscript is interesting and important. The analysis is sound. The only question is that Chapter 5 seems irrelevant to the rest of the paper. The analysis of the experimental results is also missing. If simulation is of interest, the reviewer would suggest conduct some experiments which reflect the results obtained previously about robustness and consistency.

Confidence in this Review

1-Less confident (might not have understood significant parts)


Reviewer 4

Summary

This paper examines the statistical property of the “robust k-means clustering” procedure. Specifically, they look at its robustness in terms of the finite sample breakdown point and its consistency. The idea is to use techniques from variational analysis to study the functional properties of the RKM. In particular, the paper studies the “breakdown point” of a family of robust k-means objective, with different regularization terms (f_lambda). Their main result is that this family of k-means objective is actually not robust at the presence of at most two outliers (Thm 1). This result shows that “robust” k-means objectives are actually not robust on general datasets. However, they explained that when the dataset has a clusterable structure, the objective is robust to outliers. Finally, they proved consistency of the objective.

Qualitative Assessment

Technical quality: I want to mainly point out one thing: the experiment section seems to be a little detached from the rest of the paper. The paper aims to study the property of the robust k-means objective, while the experiments compare two different robust methods for clustering, one is based on a heuristic (coordinate descent) that locally decreases the robust k-means objective, and the other is not related to any specific objective, if I understand it correctly. Novelty/Originality: I think the novelty of this paper lies in the fact that they apply techniques from optimization to study the properties of robust k-means objective. Potential impact & Usefulness: I think it is important to know that in general, the robust k-means is not robust to outliers, and that one needs to focus on dataset with more structure to obtain robustness. This discovery is inline with other recent work on the importance of clusterability for studying clustering procedures. I think the fact that they introduce the method of variational analysis to study clustering objectives could also inspire more work on understanding clustering objectives via this perspective. Clarity and presentation The interpretability of their result could be improved: in Section 3.3, it’s not clear to me what does Corollary 2 have to do with the conclusion at the end of this section. Also, it is also not clear to me what is the contribution of this paper regarding the discussion here: Is it already addressed in [2]? If not, how does the paper add insight to robustness of k-means objective with an underlying clusterable structure? In general, it'd be nice if the authors can put the work in context of related work.

Confidence in this Review

2-Confident (read it all; understood it all reasonably well)


Reviewer 5

Summary

The authors consider a variant of k-means called the robust k-means. They investigate its theoretical properties, in particular its robustness and consistency. They show that two outliers are enough to breakdown this clustering procedure. However, if the data is ''well-structured'' in the way defined by the authors, then this algorithm can recover the cluster structure in spite of the outliers. They also show the asymptotic properties of the modified robust k-means method.

Qualitative Assessment

The paper has a very good technical level nad is written in an interesting fashion (I didn't see any grammar errors). The problems raised in the paper are clearly stated. The authors show the weakness of existing robust k-means method but they also show how to fix this problem. The paper contains intersting theoretical results.

Confidence in this Review

1-Less confident (might not have understood significant parts)


Reviewer 6

Summary

The authors study the robust k-means algorithm, which is the traditional k-means algorithm with gives the points error terms, and penalizes the errors to encourage most errors to be zero. They show that adversarially moving two points is enough to break down the RKM algorithm, in the sense that the estimates for the cluster centers become arbitrarily bad. Then they show a condition that the dataset is well-structured allows the algorithm to be robust to noise. The well-structured condition is that at least half of the points can be partitioned into k clusters with sizes upper and lower bounded, and the distance between pairs of centers are lower bounded. They also show that robust k-means has nearly all of the same consistency results as the traditional k-means. They also show optimality conditions for the RKM minimization problem. Finally, they show experiments in which robust k-means performs slightly better than a variant, trimmed k-means, for several datasets.

Qualitative Assessment

This paper studies an important question, as the k-means problem is used often in practice. The robust k-means algorithm is a candidate for outperforming traditional k-means in certain settings, as it has nearly the same computational simplicity. Therefore, a theoretical analysis of robust k-means is worthwhile. The authors give two results on the robustness of this variant. They show a worst-case lower bound, and an upper bound under well-structured conditions. The conditions are a bit high, but these conditions have been studied in a previous ICML paper. The consistency result for robust k-means is interesting, and provides further justification for this variant.

Confidence in this Review

1-Less confident (might not have understood significant parts)